# Retrospective In-Context Learning for Temporal Credit Assignment with Large Language Models

**Wen-Tse Chen**[1]    **Jiayu Chen**[2]    **Fahim Tajwar**[1]    **Hao Zhu**[3]    **Xintong Duan**[1]
**Ruslan Salakhutdinov**[1]    **Jeff Schneider**[1]
[1] Carnegie Mellon University    [2] The University of Hong Kong    [3] Stanford University

## Abstract

Learning from self-sampled data and sparse environmental feedback remains a fundamental challenge in training self-evolving agents. Temporal credit assignment mitigates this issue by transforming sparse feedback into dense supervision signals. However, previous approaches typically depend on learning task-specific value functions for credit assignment, which suffer from poor sample efficiency and limited generalization. In this work, we propose to leverage pretrained knowledge from large language models (LLMs) to transform sparse rewards into dense training signals (i.e., the advantage function) through retrospective in-context learning (RICL). We further propose an online learning framework, RICOL, which iteratively refines the policy based on the credit assignment results from RICL. We empirically demonstrate that RICL can accurately estimate the advantage function with limited samples and effectively identify critical states in the environment for temporal credit assignment. Extended evaluation on four BabyAI scenarios show that RICOL achieves comparable convergent performance with traditional online RL algorithms with significantly higher sample efficiency. Our findings highlight the potential of leveraging LLMs for temporal credit assignment, paving the way for more sample-efficient and generalizable RL paradigms.

## 1   Introduction

Online learning with LLMs, guided by self-generated data and environmental feedback, offers a promising research direction for advancing beyond the constraints of demonstration [Dong et al., 2024]. However, this task is inherently difficult because valuable environmental feedback is often sparse [Andrychowicz et al., 2017, Sukhbaatar et al., 2017]. This challenge is particularly evident in multi-turn settings, where agents must execute a sequence of correct actions to receive a reward signal. Sparse environmental feedback increases both the sample complexity and the instability of the learning process [Chaudhari et al., 2024, Cao et al., 2024]. In this work, we explore effective temporal credit assignment methods to convert sparse feedback into dense training signals based on a creative use of LLMs, enabling the identification of critical states and actions in the environment and facilitating more efficient online learning (of LLM agents).

In particular, we adopt LLMs as policies and update them using a novel Retrospective In-Context Learning (RICL) algorithm. RICL is designed specifically for multi-turn setting. At each turn (i.e., environment time step), RICL first analyze environmental feedback (i.e., reward signals) by leveraging the pretrained knowledge of an LLM reflector, then incorporate this analysis into the prompt to perform an in-context update of the LLM policy. Further, we introduce a novel paradigm to estimate the advantage function of an LLM policy by utilizing the log-probabilities of both the original policy and its in-context updated version. This process is theoretically grounded, and our experiments demonstrate that it can accurately estimate the advantage function with high sample efficiency. **In this way, we transform sparse reward signals into advantage functions, which are**

**dense training signals facilitating both temporal credit assignment and policy training.** These improvements stem from the pretrained knowledge in LLMs and our new approach to estimating advantage functions using LLMs.

Further, we introduce RICOL, a novel online learning framework for LLMs that iteratively refines the policy based on the credit assignment results from RICL. Leveraging the accurate credit assignment performed by RICL and the strong generalization ability of LLMs, RICOL is significantly more sample-efficient than traditional online RL methods across multiple language-conditioned, sparse-reward sequential decision-making tasks. This demonstrates the potential of our algorithm to enable LLMs to self-improve through online learning.

In summary, our main contributions are as follows: (1) We propose RICL for temporal credit assignment, which transforms sparse reward signals into advantage functions by comparing the log-probabilities of LLM policies before and after an in-context update. (2) We propose RICOL, an online learning framework that uses advantage weighted regression to incorporate credit assignment results (i.e., advantage functions) from RICL into the LLMs' parameters. This integration enables LLM agents to self-improve based on sparse environmental feedback. (3) Empirical results demonstrate the effectiveness of RICL in temporal credit assignment and the efficacy and efficiency of RICOL in iterative policy improvement. Altogether, RICL and RICOL represent a more sample-efficient and generalizable RL paradigm for LLM agents.

## 2   Related Works

**Intrinsic Self-Correction**: In recent studies, LLMs have demonstrated advanced capabilities in self-verifying generated responses and correcting potential issues through in-context learning [Madaan et al., 2024, Kim et al., 2023]. However, critiques have suggested that intrinsic self-correction, in the absence of ground truth environment feedback, may not always enhance and could potentially degrade performance of LLMs [Huang et al., 2023, Olausson et al., 2023, Valmeekam et al., 2023]. In contrast, our focus is on learning from environmental feedback, a challenging task due to its sparse and complex nature.

**Self-Correction based on an Extra Reflector Network**: Another line of research aims to improve the self-correction capability of LLMs by fine-tuning it on an external dataset [Yao et al., 2023]. It often requires to collect an additional dataset especially tailored for the training of a reflector network. For instance, Chen et al. [2024] design an erroneous solution rewriting task for reflector training. In contrast, we perform self-correction using free-form environmental feedback and do not need to train an extra network.

**Self-Correction based on Induction**: Prior work has explored encoding past experiences into text-form memory and incorporating this memory into prompts to enhance decision-making in subsequent trials [Shinn et al., 2024, Zhao et al., 2024, Zelikman et al., 2022]. There is also a line of works leveraging LLMs to generate rewards – either as codes or natural language prompts – and iteratively refine these rewards based on environmental feedback through in-context learning [Ma et al., 2023, Kwon et al., 2023, Yu et al., 2023, Li et al., 2024]. These studies assume that LLMs, through in-context learning, can infer environmental rules and adapt to new trajectories/experiences with only a few observed transitions. In contrast to these approaches, we assume that the rewards or reflections obtained from one trajectory are not directly transferable to other trajectories. Thus, in this work, we proposed to use in-context learning in a "retrospective" manner.

**Self-Correction Through Preference Alignment**: Iterative RPO [Pang et al., 2024] labels preference data based on environmental feedback and uses this feedback as a supervised signal to fine-tune LLMs. RICO-GRPO [Wang et al., 2025] use trajectory level reward and ground normalization to estimate the advantage function for multi-turn online RL training. However, these methods don't perform explicit temporal credit assignment. In contrast, our approach generates dense rewards by leveraging the pretrained knowledge of LLMs through in-context learning. This enables the policy to be trained with more informative supervision signals.

**Temporal Credit Assignment using LLMs**: The goal of temporal credit assignment is to identify the critical states and actions in sequential decision-making. A common approach to credit assignment involves estimating a value or advantage function based on learning experiences [Schulman et al., 2015, Ahmadian et al., 2024]. Unlike RICO-PPO [Wang et al., 2025] training a value network from

scratch, which can be sample inefficient, our approach leverages in-context learning with pretrained LLMs to perform credit assignment more effectively. **Please refer to Table 1 for a more intuitive comparison of our proposed method with related works on self-correction in LLMs.**

# 3    Preliminary

**Markov Decision Process:** A Markov Decision Process (MDP) can be described as a tuple $(\mathcal{S}, \mathcal{A}, P, r, \gamma, \rho_0)$. Here, $\mathcal{S}$ and $\mathcal{A}$ represent the state and action space, respectively; $P : \mathcal{S} \times \mathcal{A} \times \mathcal{S} \to [0, 1]$ is the transition kernel; $r : \mathcal{S} \times \mathcal{A} \to \mathbb{R}$ is a reward function; $\gamma \in [0, 1)$ is the discount factor; $\rho_0(\cdot)$ denotes a distribution of the initial state.

**KL-Regularized Policy Updates:** KL-regularized policy optimization [Abdolmaleki et al., 2018] enhances policy stability by constraining each update to remain close to the previous policy. The objective is to learn a new policy $\pi_{k+1}$ that maximizes the expected advantage while penalizing deviations from the old policy: $\pi_{k+1} = \arg\max_\pi \mathbb{E}_{s \sim \rho^{\pi_k}, a \sim \pi(\cdot|s)}[A^{\pi_k}(s, a) - \beta D_{KL}(\pi(\cdot|s)||\pi_k(\cdot|s))]$, where $A^{\pi_k}(s, a)$ is the advantage function, and $\beta > 0$ controls the trade-off between policy improvement and stability. The resulting policy update admits a closed-form solution:

$$\pi_{k+1}(a|s) \propto \pi_k(a|s) \exp\left(\tfrac{1}{\beta} A^{\pi_k}(s, a)\right), \quad \forall (s, a) \in \mathcal{S} \times \mathcal{A}. \tag{1}$$

This update exponentially favors advantageous actions while ensuring smooth policy changes via the KL constraint.

# 4    Retrospective In-Context Learning for Temporal Credit Assignment

Given the current policy and its experience, the goal of temporal credit assignment is to identify critical state-action pairs for sequential decision-making. Critical states are those where policy decisions significantly influence the expected return-to-go, while critical actions indicate policy update directions at those states. In RL, credit assignment is typically performed by estimating the advantage function, which measures the impact of different actions at a given state on the future return-to-go under the current policy. **Thus, in this section, we propose leveraging LLMs to estimate advantage functions for temporal credit assignment, using retrospective in-context learning guided by environmental feedback.**

## 4.1    LLMs as Policies

Consider the case where an LLM is used as a policy $\pi_0$[1]. This policy can be refined through in-context learning by embedding task descriptions or goal-related hints into the prompt, resulting in an in-context updated policy, $\pi'$. Unlike standard RL, the transition from $\pi_0$ to $\pi'$ in in-context learning is not an explicit optimization process. However, we can infer the implicit advantage function driving this policy improvement by analyzing the log-probabilities of $\pi_0$ and $\pi'$. The theoretical foundation of this approach is established in the following theorem.

**Theorem 4.1.** *Let $\pi_0 : \mathcal{S} \times \mathcal{A} \to (0, 1)$ and $\pi' : \mathcal{S} \times \mathcal{A} \to (0, 1)$ be any two policies in an MDP with transition kernel $P : \mathcal{S} \times \mathcal{A} \times \mathcal{S} \to [0, 1]$ and discount factor $\gamma \in [0, 1]$. Then, there exists a reward function $r : \mathcal{S} \times \mathcal{A} \to \mathbb{R}$ such that the following relationship holds:*

$$\beta \log \frac{\pi'(a|s)}{\pi_0(a|s)} \propto A_r^{\pi_0}(s, a), \tag{2}$$

*where $\beta > 0$ is a known scaling parameter, and $A_r^{\pi_0}(s, a)$ denotes the advantage function under policy $\pi_0$ in the finite MDP $(\mathcal{S}, \mathcal{A}, P, r, \gamma)$.*

*Proof. See Appendix B.*

Our algorithm, as detailed later, assumes the existence of a reward function such that the corresponding advantage function explains the discrepancy between the in-context updated policy $\pi'$ and the actor policy $\pi_0$. Theorem 4.1 establishes the existence of such a reward function for any pair of policies, thereby supporting the soundness of our proposed online RL framework.

---

[1]For an LLM-based policy, each state $s$ corresponds to a prompt, while each action $a$ is represented as a sequence of tokens (e.g., "turn right" or "move forward"). Consequently, the probability $\pi(a|s)$ can be computed as the token prediction probability of $a$ occurring after $s$ in the LLM's autoregressive generation.

## 4.2 Implementation Details

The procedure for estimating the advantage function $A_r^{\pi_0}(s, a)$ of a policy $\pi_0$ is as follows: First, we collect $n$ trajectories starting from the state-action pair $(s, a)$, by executing the policy $\pi_0$. Then, for each trajectory $\tau^{(i)}$, we perform retrospective in-context learning (introduced in the following subsection) to update the LLM and get an updated policy $\pi'^{(i)}$.

According to Theorem 4.1, we have $\beta \log \pi'^{(i)}(a|s) = \beta \log \pi_0(a|s) + A_{r_i}^{\pi_0}(s, a) - \beta \log Z^{(i)}(s)$, where $Z^{(i)}(s)$ is a partition function to ensure that $\pi'^{(i)}$ is a valid probability distribution. In this paper, we consider only tasks with a discrete and enumerable action space, allowing $Z^{(i)}(s)$ to be computed by summing over the probabilities of all possible actions. $A_{r_i}^{\pi_0}(s, a)$ is derived from $\log \pi'^{(i)}(a|s)$ and $\log \pi_0(a|s)$ and is a sample estimate for the ground-truth advantage function $A_{r^*}^{\pi_0}(s, a)$. Thus, we can use the sample mean $\bar{A}_r^{\pi_0}$ as the estimated advantage function, which is defined as:

$$\bar{A}_r^{\pi_0}(s, a) = \frac{\beta}{n} \sum_{i=1}^{n} \left( \log \frac{\pi'^{(i)}(a|s)}{\pi_0(a|s)} + \log Z^{(i)}(s) \right). \tag{3}$$

Empirical results demonstrate that $\bar{A}_r^{\pi_0}(s, a)$ closely approximates the ground-truth advantage function $A_{r^*}^{\pi}(s, a)$. **Recall the policy improvement step shown in Eq. (1), this alignment suggests that the in-context learning with LLMs implicitly performs KL-Regularized Policy Updates.**

## 4.3 Retrospective In-Context Learning

As aforementioned, $\pi'$ is updated from $\pi_0$ using Retrospective In-Context Learning (RICL), a novel in-context learning algorithm for refining LLM agents. RICL is illustrated as step ② and step ③ in Figure 1. First, given a state $s_t$ and its corresponding hindsight trajectory – a sequence of future states, actions, and rewards $\{s_{t:T}, a_{t:T-1}, r_{t:T-1}\}$ where $T$ denotes the episode horizon –we input this trajectory into a reflector LLM, $\pi_{reflect}$, to generate a verbal feedback $f_t$. The reflector can be any LLM, prompted to analyze the hindsight trajectory and provide corrective feedback to improve the agent's performance at state $s_t$. Next, we integrate the feedback $f_t$ into the original LLM's prompt, yielding an in-context updated policy $\pi'(\cdot|s_t) = \pi_0(\cdot|s_t, f_t)$. This policy is then used to estimate the advantage function for temporal credit assignment, as introduced in the previous subsection.

RICL offers two key advantages over previous in-context learning algorithms [Shinn et al., 2024]. First, it generates verbal feedback for each action individually, whereas prior methods provide a single feedback for the entire trajectory, allowing for more fine-grained guidance. Second, it employs retrospective updates, meaning the policy is updated specifically for the states it has just encountered, rather than requiring the reflector LLM to generalize to unseen states. This method reduces the complexity of tasks assigned to in-context learning and lessens the dependence on reflector LLMs.

# 5 Retrospective In-Context Online Learning

Once the log-probabilities of the in-context updated policy are obtained, which indicate the advantage function, policy improvement can be performed straightforwardly using advantage-weighted regression [Peng et al., 2019]. In this section, we integrate the policy improvement phase with the previously introduced policy evaluation phase (i.e., RICL) to develop a practical online RL algorithm: Retrospective In-Context Online Learning (RICOL).

## 5.1 Policy Improvement based on RICL

To learn from the in-context improved policy $\pi'$, we can adopt the following objective:

$$\pi^* = \underset{\pi}{\arg\max} \, \mathbb{E}_{s, a \sim \pi} \left[ A^{\pi_0}(s, a) \right] = \underset{\pi}{\arg\max} \, \mathbb{E}_{s \sim d_\pi, a \sim \pi(\cdot|s)} \left[ \log \pi'(a|s) - \log \pi_0(a|s) \right], \tag{4}$$

where $d_\pi(s) = (1 - \gamma) \sum_{t=0}^{\infty} \gamma^t \rho_t^\pi(s)$ represents the discounted state occupancy measure for policy $\pi$ and $\rho_t^\pi(s)$ denotes the probability that policy $\pi$ visits state $s$ at time $t$.

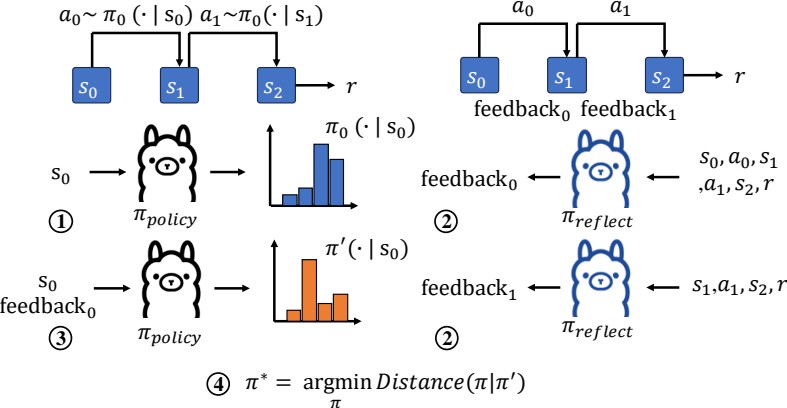

Figure 1: The pipeline of retrospective in-context online learning, where step ② and step ③ represent retrospective in-context learning.

Inspired by AWR [Peng et al., 2019], we can reuse samples collected with $\pi_0$ to avoid resampling from $\pi$ and so improve the algorithm's sample efficiency, with a smarter objective design:

$$\min_\pi \mathbb{E}_{s \sim d_{\pi_0}} \left[ D_{KL} \left( \frac{1}{Z(s)} \odot \exp\left( (1-\alpha) \log \pi_0(\cdot|s) + \alpha \log \pi'(\cdot|s) \right) \| \pi(\cdot|s) \right) \right] \quad (5)$$

where $Z(s)$ is a partition function , $\odot$ denotes element-wise multiplication, and hyperparameter $\alpha$ controls the size of the trust region. The derivation from Eq. (4) to Eq. (5) is detailed in Appendix A.

Integrating the sampling policy $\pi_0$ into the objective function introduces trust region constraints, akin to those used in online RL [Schulman, 2015]. These constraints regulate policy updates, ensuring that deviations from the current sampling policy remain bounded during optimization. By anchoring updates within a local region around $\pi_0$, the trust region term helps mitigate overfitting to potentially noisy verbal feedback generated through in-context learning.

## 5.2 Pipeline of RICOL

**Trajectory Collection:** During iteration $k$, the policy $\pi_k$ interacts with the environment to collect a batch of trajectories $\tau_k$. Each trajectory comprises state-action-reward sequences $(s_{1:T}, a_{1:T-1}, r_{1:T-1})$.

**Policy Evaluation:** For each state $s_t$ in $\tau_k^{(i)}$, RICOL applies RICL (detailed in Section 4.3) to perform an in-context update on the policy $\pi_k(\cdot|s_t)$, resulting in an improved policy $\pi_{k+1}'^{(i)}(\cdot|s_t)$. To estimate $\pi_{k+1}'(\cdot|s_t)$, we follow the process: $[\pi_{k+1}'^{(i)}(\cdot|s_t), \pi_k(\cdot|s_t)]_{i=1}^n \rightarrow [A_{r_i}^{\pi_k}(s_t, \cdot)]_{i=1}^n \rightarrow \bar{A}_r^{\pi_k}(s_t, \cdot) \rightarrow \pi_{k+1}'(\cdot|s_t)$, where the first two steps are detailed in Section 4.2 and the last step is according to $\beta \log \frac{\pi_{k+1}'(a_t|s_t)}{\pi_k(a_k|s_t)} \propto \bar{A}_r^{\pi_k}(s_t, a_t)$ (i.e., Eq. (2)).

---

**Algorithm 1** RICOL

**Input:** $\pi_0, \pi_{reflect}, K, n$
**for** $k = 0 \cdots K - 1$ **do**
  **for** $i = 1 \cdots n$ **do**
    $\tau_k^{(i)} \sim \pi_k(\cdot|s_0)$
    **for** $s_t = s_0 \cdots s_{T-1} \in \tau_k^{(i)}$ **do**
      $\text{feedback}_t \sim \pi_{reflect}(\cdot|s_{t:T}, r_{t:T-1})$
      $\pi_{k+1}'^{(i)}(\cdot|s_t) \leftarrow \pi_k(\cdot|s_t, \text{feedback}_t)$
    **end for**
  **end for**
  $\pi_{k+1} \leftarrow \text{argmin}_\pi \mathbb{E}_{s \sim \tau_k}[D_{KL}(\frac{1}{Z(s)} \odot \exp(​$
  $(1-\alpha) \log \pi_k(\cdot|s) + \alpha \log \pi'(\cdot|s))\|\pi(\cdot|s))]$
**end for**

---

**Policy Improvement:** After computing $\pi_{k+1}'(\cdot|s_t)$ across the batch of states in $\tau_k$, the policy $\pi_k$ is updated via gradient descent to minimize the objective in Eq. (5). For environments with small, discrete action spaces, the KL-divergence in Eq. (5) is computed exactly by enumerating all actions. This requires querying the LLM $|\mathcal{A}|$ times per state. For general tasks, the KL-divergence can be estimated by sampling from $\pi^*(\cdot|s) \propto \exp\left( (1-\alpha) \log \pi_k(\cdot|s) + \alpha \log \pi'(\cdot|s) \right)$, avoiding exhaustive action enumeration.

# 6 Experiments

In this section, we present a series of experiments to address the following research questions (RQs): **RQ1**: How effective is RICL at credit assignment in multi-turn tasks? Compared to traditional RL methods, how sample-efficient is RICL in credit assignment? **RQ2**: The proposed method relies on LLMs, which may be unreliable. How does RICL compare to other reflection-based LLM methods in terms of reliability? Given that RICL may be unreliable, can RICOL still learn effectively from noisy labels? **RQ3**: Can RICOL efficiently learn from sparse rewards in multi-turn tasks? How does its performance compare to baseline methods, and what types of tasks is it best suited for?

## 6.1 Environments

We evaluate our algorithms on a 1D Key-Door environment and four BabyAI scenarios. In all settings, both the policy inputs (observations) and outputs (actions) are represented in text form. The action space is discrete and enumerable. In the 1D Key-Door environment, the agent operates on a 1D grid world, moving along the x-axis. The key is placed in the leftmost cell, and the door in the rightmost. The agent starts without the key and must first move left to retrieve it, then move right to unlock the door. This setup enables exact computation of the ground-truth advantage at each cell, making it ideal for evaluating RICL's credit assignment capabilities. We test RICOL on four BabyAI [Carta et al., 2023] tasks: *goto*, *pickup*, *pick_up_seq_go_to*, and *open*. BabyAI is a 2D grid world where the agent must complete language-conditioned tasks. These tasks are challenging to LLMs, due to their limited spatial reasoning abilities and the long episode lengths. As shown in Figure 4, even GPT-4o mini achieves a success rate of no more than 40% across all four BabyAI scenarios. More details on the tasks and associated prompt templates are available in Appendix E.

## 6.2 Sample Efficiency of RICL in Credit Assignment (RQ1)

In this subsection, we demonstrate that RICL achieves greater sample efficiency ($100\times$) compared to classic Monte Carlo (MC) methods when estimating the advantage function in the 1D Key-Door scenario. Since RL methods alternate between policy evaluation (value function estimation) and policy improvement, a more sample-efficient policy evaluation approach can significantly speed up the overall training process.

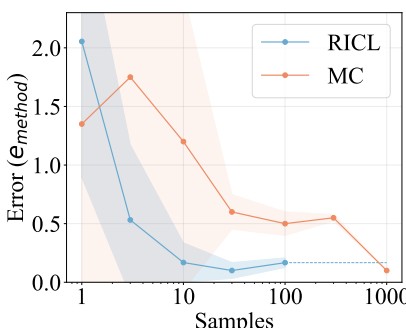

Figure 2: Comparison of error in advantage function estimation. The x-axis represents the number of trajectories used for estimation, while the y-axis shows the mean error between the estimated advantage and the ground-truth advantages. RICL achieves significantly lower error with fewer samples (around 10) compared to Monte Carlo, which requires about 1000 samples for similar accuracy. Additionally, the error of RICL is more stable across trials (lower variance).

We measure the discrepancy between estimated and ground-truth advantage functions for both MC and RICL. For each $(s_t, a_t)$, we generate $n$ trajectories starting from $s_t$ using the policy $\pi_0$. MC estimates the Q-value $\hat{Q}^{\pi_0}(s_t, a_t)$ as the average return-to-go across the $n$ trajectories. The estimated advantage function is defined as $\hat{A}^{\pi_0}(s_t, a_t) = \hat{Q}^{\pi_0}(s_t, a_t) - \hat{V}^{\pi_0}(s_t)$, where $\hat{V}^{\pi_0}(s_t) = \mathbb{E}_{a \sim \pi_0(\cdot|s_t)}[\hat{Q}^{\pi_0}(s_t, a_t)]$. The updated policy under MC is accordingly defined as $\pi_{\text{MC}}(a_t|s_t) \propto \pi_0(a_t|s_t) \exp(\hat{A}^{\pi_0}(s_t, a_t)/\beta)$. On the other hand, RICL estimates the advantage function $\bar{A}^{\pi_0}(s_t, a_t)$ directly from $n$ trajectories using the procedure outlined in Section 4.2. The corresponding policy $\pi_{\text{RICL}}(a_t|s_t)$ is similarly defined as $\pi_{\text{RICL}}(a_t|s_t) \propto \pi_0(a_t|s_t) \exp(\bar{A}^{\pi_0}(s_t, a_t)/\beta)$. Finally, to evaluate both methods, we leverage ground-truth advantage function $A^{\pi_0}(s_t, a_t)$ to compute the reference policy $\pi_{\text{gt}}(a_t|s_t) \propto \pi_0(a_t|s_t) \exp(A^{\pi_0}(s_t, a_t)/\beta)$.

The performance of MC and RICL is assessed by computing the expected KL divergence between their respective policies and the ground-truth policy. This expectation is taken over states sampled from $\rho_0$, a uniform distribution over all possible initial states in the 1D grid-world. Formally, the error for each method is defined as: $e_{\text{method}} = \mathbb{E}_{s \sim \rho_0}[D_{KL}(\pi_{\text{method}}(\cdot|s) \| \pi_{\text{gt}}(\cdot|s))]$. By definition

of the policies, when $\pi_{\text{method}}(\cdot|s)$ closely approximates $\pi_{\text{gt}}(\cdot|s)$, the estimated advantage function likewise approaches the ground-truth advantage function.

For each value of $n$ (number of trajectories), we run eight trials with different random seeds. We report the mean KL divergence and standard deviation. These results are visualized in Figure 2, where the x-axis represents the number of samples $n$. As shown in Figure 2, LLMs achieve an accurate estimate of advantage functions with just 10 samples, while MC requires around 1000 samples to reach a similar accuracy. Furthermore, the standard deviation of the error of RICL is much lower than that of MC. This suggests that RICL is particularly advantageous for tasks that require high sample efficiency, where leveraging the pretrained knowledge of LLMs is crucial. **In Appendix H, we compare RICL with an LLM-based method (Retroformer [Yao et al., 2023]) in terms of sample efficiency for value function estimation. Further, in Appendix G, we show that RICL can identify critical states in sequential decision-making, based on the result of credit assignment.**

## 6.3 RICL Enables More Reliable In-Context Updates via Retrospective Design (RQ2)

Standard ICL follows these steps: (1) Use an actor LLM to generate a trajectory $\tau$; (2) Use a reflector LLM to generate verbal feedback based on $\tau$; (3) Use the verbal feedback to perform an in-context update of the actor. While integrating ICL into the training loop improves sample efficiency, it also introduces instability, because we cannot assume that experiences from one trajectory can be applied to any other state in the environment. As an improvement, RICL shares steps (1)–(3) with ICL, but uses the in-context updated policy to compute the action distribution over **the states in $\tau$ only**. In this section, we empirically demonstrate that using environmental feedback to retrospectively in-context update the policy on the same trajectory that produced the feedback (i.e., RICL) leads to more effective and reliable policy improvement than updating the policy on a different trajectory (i.e., ICL).

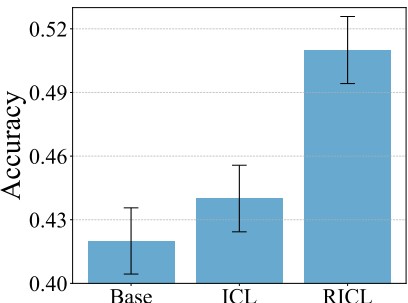

Figure 3: Accuracy comparison of ICL and RICL on predicting expert actions in the BabyAI *goto* scenario across 1000 trajectories. The Base bar shows the zero-shot performance of LLaMA-3.1-8B-Instruct. RICL outperforms ICL by 7.2%, demonstrating the effectiveness of retrospective updates.

Figure 3 compares RICL and ICL in the BabyAI *goto* scenario and uses LLaMA-3.1-8B-Instruct as actor and reflector LLMs. We conduct experiments on 1000 different trajectories. We use accuracy as the evaluation metric, defined as the probability that the policy selects the expert action. The base policy reflects the performance of the zero-shot LLaMA-3.1-8B-Instruct model. In terms of accuracy, RICL outperforms ICL by 7.2%, confirming that retrospective updates are both more effective and more reliable.

## 6.4 Benchmarking RICOL (RQ3)

Here, we compare RICOL against four baseline algorithms across four BabyAI scenarios and demonstrate that it achieves state-of-the-art performance in solving multi-turn tasks with higher sample efficiency. We adopt Llama-3.2-3B-Instruct as the policy and GPT-4o mini as the reflector. Regarding evaluation metrics, we use environment time steps (for sampling) and task success rates to evaluate sample efficiency and effectiveness of the algorithms, respectively. The remaining experiments in the paper report the mean and standard deviation across three different random seeds.

**Baselines: (1) GPT-4o mini:** This baseline uses GPT-4o-mini as the policy, providing a measure of the zero-shot performance of state-of-the-art LLMs on the benchmark tasks. Moreover, since GPT-4o-mini serves as the reflector in our algorithm, improvements over this baseline demonstrate that simple behavioral cloning of the reflector is insufficient, highlighting the necessity of our algorithmic design for effective policy improvement. **(2) Reflexion:** Reflexion [Shinn et al., 2024] relies on in-context learning with iterative self-reflection. A base policy first interacts with the environment to collect a trajectory, which is used to prompt an LLM for verbal feedback. The policy then uses this feedback to generate a new trajectory, which is again used to update the feedback. This process repeats iteratively. **(3) PPO (3B):** We fine-tune a Qwen2.5-3B-Instruct model using PPO, where the 3B model serves as both the policy's and the critic's backbone. More details are provided in

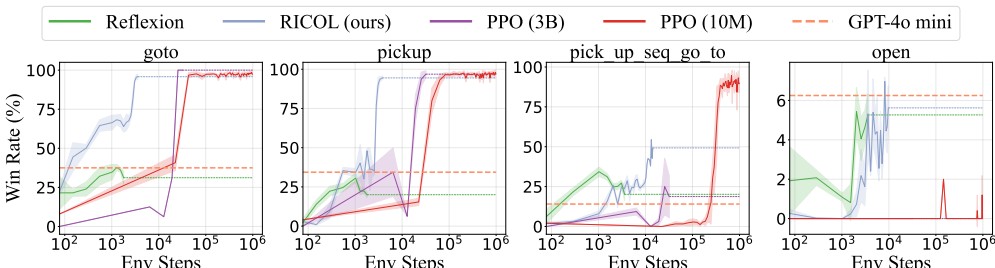

Figure 4: Comparison of our method (RICOL) against four baseline algorithms across four BabyAI scenarios. RICOL consistently demonstrates superior sample efficiency, achieving strong performance with significantly fewer interactions. Notably, RICOL outperforms both PPO (10M) and PPO (3B), by over $50\times$ and $10\times$ fewer environment steps, respectively. Compared to Reflexion, an in-context learning method using trajectory-level verbal feedback, RICOL exhibits better convergent performance by leveraging temporal credit assignment (from RICL) and state-specific feedback. Additionally, RICOL surpasses GPT-4o mini, despite using a smaller policy model (LLaMA-3.2-3B-Instruct), underscoring the importance of interactive learning from the environment. **As a useful trick to boost performance, we use the real environment rewards as the advantage and apply advantage-weighted regression during the second stage of training, after RICOL completes its predefined training schedule in the first stage.**

Appendix F. We use Qwen instead of LLaMA due to prior findings [Gandhi et al., 2025] suggesting Qwen exhibits stronger reasoning behaviors such as verification and backtracking. **(4) PPO (10M):** This baseline uses two randomly initialized 3-layer MLPs for the policy and critic, respectively. It serves to test whether LLMs' pretrained knowledge provides any advantage during learning.

**Results: (1)** Figure 4 shows that the in-context learning method *Reflexion* is sample-efficient, but its performance gains saturate quickly. We attribute this to two factors. First, unlike other algorithms, in-context learning does not update the LLM's parameters, limiting its ability to effectively acquire new knowledge. Second, *Reflexion* relies on verbal feedback at the trajectory level, while our method provides verbal feedback for each specific state. As a result, *Reflexion* saturates early because the general (trajectory-level) rules it can easily capture from the environment are limited. Here is an example of the verbal feedback learned by *Reflexion*: "In future navigation tasks, look for opportunities to reposition yourself to move closer to your goal". In contrast, our method generates more precise, state-specific feedback: "Prioritize picking up the green key first by taking action D when you are in front of it, rather than moving towards the grey box." **(2)** Regarding comparisons with PPO-based methods, PPO (10M) requires approximately 0.1 to 1 million time steps to converge across most scenarios, highlighting the sample inefficiency of training from scratch without leveraging LLMs' pretrained knowledge. In contrast, our method achieves comparable performance while using nearly 50 times less data. PPO (3B), which adopts LLMs as both the policy and critic, converges in about 20,000 time steps on most scenarios. Our method is approximately $10\times$ **more sample efficient** than this approach. While our algorithm also uses an LLM for the policy, it does not require training a critic. Instead, our value function approximation method, RICL, is more sample-efficient than Monte Carlo-based ones (as used in PPO), further supporting the argument made in Section 6.2. **(3)** Finally, our method outperforms GPT-4o-mini in success rates across all four scenarios, despite using a smaller 3B model as the policy. This underscores the importance of learning through interaction with the environment, rather than merely imitating the outputs of a larger models. In addition to sample efficiency, we also report the training time of each algorithm on each task in Table 2. In summary, RICOL is well-suited for settings with limited simulation budgets, typically ranging from 1,000 to 10,000 environment steps. In these scenarios, where environment interactions are costly, sample efficiency is crucial, making our method a compelling and practical choice.

## 6.5   RICOL Benefits from Credit Assignment (RQ3)

RICOL can be interpreted as a combination of RICL for credit assignment and AWR for policy improvement. In contrast, RWR [Peters and Schaal, 2007] directly uses trajectory-level rewards as returns to perform AWR updates on an LLM policy. Unlike RICOL, RWR does not assign credit to individual time steps within an episode, making it less effective for tasks with sparse rewards. Here, we compare RWR with RICOL to show that RICOL benefits from credit assignment.

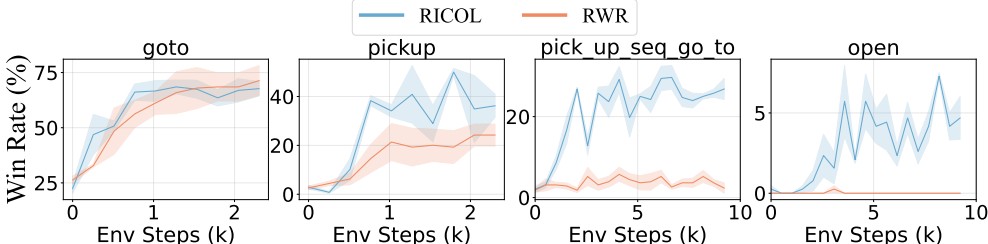

Figure 5: RICOL employs in-context credit assignment to generate dense learning signals, enabling more sample-efficient policy training. In contrast, RWR lacks credit assignment, depends on strong base policies with high initial success rates, and performs poorly on tasks with sparse rewards.

Figure 5 shows that RWR achieves competitive performance only in the *goto* scenario, where the base policy already attains a relatively high success rate of 25%. This highlights the limitation of relying solely on trajectory-level rewards. In contrast, our method leverages RICL to generate dense supervised signals, as described in Section 4, thereby enabling more effective policy training for LLMs, particularly in sparse-reward environments.

## 6.6 RICOL is Robust to Noisy Verbal Feedback (RQ2)

Figure 6 illustrates how the accuracy of verbal feedback affects the performance of RICOL in the BabyAI *goto* scenario. In this experiment, we train the agent using hand-crafted verbal feedback and systematically vary its accuracy to assess RICOL's robustness. The hand-crafted verbal feedback acts as a binary correctness indicator for the actions in trajectories, framed as: "In the previous attempt, I chose action [Action]. This time, I will maintain/change the selected action." To manipulate feedback accuracy, we randomly flip the correctness labels in the verbal feedback at different rates, thus simulating various levels of feedback reliability. An accuracy of 100% corresponds to perfectly accurate feedback, while random feedback yields an accuracy of approximately 50%. As shown in Figure 6, RICOL remains effective even when verbal feedback accuracy drops to 70%. This result demonstrates that RICOL does not rely heavily on accurate verbal feedback during training. We attribute this robustness to the trust region term introduced in the loss function during the policy improvement step (Eq. 5), which helps normalize the impact of noisy supervision and stabilize learning.

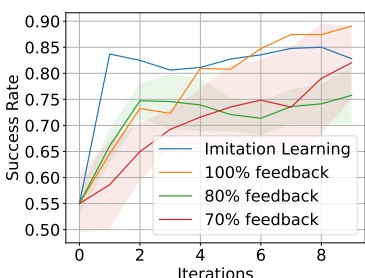

Figure 6: The performance of RICOL under varying verbal feedback accuracy. The agent is trained with hand-crafted verbal feedback of different accuracy levels. Despite the presence of noise, RICOL maintains strong performance. Note that 50% accuracy corresponds to random feedback.

## 7   Conclusion

We propose a novel in-context learning algorithm, RICL, to convert sparse environmental feedback into dense training signals for estimating advantage functions and conducting temporal credit assignment. We further propose an online learning framework, RICOL, for iterative policy improvement based on the RICL results. Our method mitigates the instability of in-context learning and enables continuous policy refinement. Empirical results on the 1D Key-Door scenario demonstrate RICL's effectiveness in credit assignment, while results on the BabyAI benchmark show that RICOL is significantly more sample-efficient than traditional online RL methods. As a future direction, we aim to extend and test our algorithm on token-level MDPs and reasoning tasks.

**Limitations**: RICOL only supports tasks with discrete, finite action spaces, as when calculating the KL divergence in Eq. 5, it needs to calculate the partition term $Z$ by enumerating all the actions. We believe this limitation is acceptable, as we use LLMs as policies and their output space, i.e., the token space, is inherently discrete. When the action space is not enumerable, for example, when the policy generates extremely long chains of thought before producing the next action, we can replace action enumeration with action sampling. We leave this extension for future work.

# 8 Acknowledgement

This work was supported in part by the U.S. Army Futures Command under Contract No. W519TC-23-C-0030.

This work used Delta at the National Center for Supercomputing Applications (NCSA) through allocation CIS250651 from the Advanced Cyberinfrastructure Coordination Ecosystem: Services & Support (ACCESS) program [Boerner et al., 2023], which is supported by U.S. National Science Foundation grants #2138259, #2138286, #2138307, #2137603, and #2138296.

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

# A Derivation of the Loss Function

The derivation is mainly from Peng et al. [2019]. The goal of the gradient update step is to bring $\pi$ closer to $\pi'$. One way to achieve this is by maximizing the following objective:

$$\eta(\pi) = \mathbb{E}_{s \sim d_\pi(\cdot), a \sim \pi(\cdot|s)} \left[ \log \pi'(a|s) - \log \pi_k(a|s) \right], \tag{6}$$

where $d_\pi(s) = \sum_{t=0}^{\infty} \gamma^t p(s_t = s|\pi)$ represents the unnormalized discounted state distribution induced by the policy $\pi$, and $p(s_t = s|\pi)$ is the likelihood of the agent being in state $s$ after following $\pi$ for $t$ time-steps. For simplicity, we use $\pi_k$ in the appendix but $\pi_0$ in the main text; both denote the sampling policy.

In practice, to enhance sample efficiency, we aim to use $d_{\pi_k}(s)$ instead of $d_\pi(s)$ when estimating $\eta(\pi)$, where $\pi_k$ is the policy used to sample new data. Therefore, similar to Peng et al. [2019], we can employ $\hat{\eta}(\pi)$ as an approximation of $\eta(\pi)$ in practical implementations.

$$\hat{\eta}(\pi) = \sum_s d_{\pi_k}(s) \sum_a \pi(a|s) \left[ \log \left( \frac{\pi'(a|s)}{\pi_k(a|s)} \right) \right]. \tag{7}$$

$\hat{\eta}(\pi)$ is a reasonable estimator of $\eta(\pi)$ only when $\pi$ and $\pi_k$ are sufficiently similar. Therefore, rather than directly solving the optimization problem $\max_\pi \eta(\pi)$, we can instead reformulate it as the following optimization problem:

$$\operatorname*{argmax}_\pi \sum_s d_{\pi_k}(s) \sum_a \pi(a|s) \left[ \log \left( \frac{\pi'(a|s)}{\pi_k(a|s)} \right) \right]$$
$$\text{s.t.} \sum_s d_{\pi_k}(s) D_{KL} \left( \pi(\cdot|s) || \pi_k(\cdot|s) \right) \le \epsilon \tag{8}$$
$$\sum_a \pi(a|s) = 1, \forall s.$$

By applying the method of Lagrange multipliers to the constrained optimization problem, the optimal policy can be expressed as follows:

$$\pi^*(a|s) = \frac{1}{Z(s)} \pi_k(a|s) \exp \left( \frac{1}{\alpha} \log \frac{\pi'(a|s)}{\pi_k(a|s)} \right), \tag{9}$$

where $Z(s) = \sum_{a'} \pi_k(a'|s) \exp \left( \frac{1}{\alpha} \log \frac{\pi'(a|s)}{\pi_k(a|s)} \right)$ normalizes the optimal policy and $\alpha$ is the Lagrange multiplier.

Finally, we need to project $\pi^*$ back onto the manifold of parameterized policies. This can be achieved by optimizing the following objective:

$$\operatorname*{argmin}_\pi \mathbb{E}_{s \sim d_{\pi_0}(s)} \left[ D_{KL} \left( \pi^*(\cdot|s) || \pi(\cdot|s) \right) \right]$$
$$= \operatorname*{argmin}_\pi \mathbb{E}_{s \sim d_{\pi_0}(s)} \left[ D_{KL} \left( \frac{1}{Z(s)} \cdot \pi_k(\cdot|s) \odot \exp(\frac{1}{\alpha} \log \frac{\pi'_{k+1}(\cdot|s)}{\pi_k(\cdot|s)}) || \pi(\cdot|s) \right) \right]$$
$$= \operatorname*{argmin}_\pi \mathbb{E}_{s \sim d_{\pi_0}(s)} \left[ D_{KL} \left( \frac{1}{Z(s)} \cdot \exp((1 - \frac{1}{\alpha}) \log \pi_k(\cdot|s) + \frac{1}{\alpha} \log \pi'_{k+1}(\cdot|s)) || \pi(\cdot|s) \right) \right]$$
$$= \operatorname*{argmin}_\pi \mathbb{E}_{s \sim d_{\pi_0}(s)} \left[ D_{KL} \left( \frac{1}{Z(s)} \cdot \exp((1 - \alpha^*) \log \pi_k(\cdot|s) + \alpha^* \log \pi'_{k+1}(\cdot|s)) || \pi(\cdot|s) \right) \right]$$
$$, \tag{10}$$

where $\odot$ represents the element-wise multiplication. In practice, we treat $\alpha^* = 1/\alpha$ as a hyperparameter that controls the size of the trust region.

# B Proof of Theorem 4.1

*Proof.* Let $\pi_0 : \mathcal{S} \times \mathcal{A} \to (0,1)$ and $\pi : \mathcal{S} \times \mathcal{A} \to (0,1)$ be any two policies in a finite MDP $(\mathcal{S}, \mathcal{A}, P, r, \gamma)$, where $r : \mathcal{S} \times \mathcal{A} \to \mathbb{R}$ is unknown. According to Eq. (2), we have: ($\forall s \in \mathcal{S}, a \in \mathcal{A}$)

$$\pi(a|s) = \frac{\pi_0(a|s) \exp\left(A_r^{\pi_0}(s,a)/\beta\right)}{Z(s)} \Rightarrow A_r^{\pi_0}(s,a) = \beta \left( \log \frac{\pi(a|s)}{\pi_0(a|s)} + \log Z(s) \right) \quad (11)$$

Here, $Z(s)$ is a normalization factor and is a function of $\pi(\cdot|s)$ and $\pi_0(\cdot|s)$ since $\sum_a A_r^{\pi_0}(s,a)/\beta = \sum_a \log \frac{\pi(a|s)}{\pi_0(a|s)} + \log Z(s) = h(\pi_0)$. Thus, given $\pi$ and $\pi_0$, we can acquire the advantage function $A_r^{\pi_0}(s,a)$ for all $(s,a)$.

Next, we derive the relation between $A_r^{\pi_0}$ and $r$ under the maximum entropy RL framework. According to Ziebart et al. [2008], Ziebart [2010], the advantage function $A_r^{\pi_0}$, value function $V^{\pi_0}$, and Q function $Q^{\pi_0}$ of a policy $\pi_0$ are related as follows: ($t$ is a time step and $(s_t, a_t) \in \mathcal{S} \times \mathcal{A}$.)

$$A_r^{\pi_0}(s_t, a_t) = Q^{\pi_0}(s_t, a_t) - V^{\pi_0}(s_t)$$

$$V^{\pi_0}(s_t) = \mathbb{E}_{a_t \sim \pi_0(\cdot|s_t)} [Q^{\pi_0}(s_t, a_t) - \beta \log \pi_0(a_t|s_t)] = \sum_{a_t} \pi_0(a_t|s_t) Q^{\pi_0}(s_t, a_t) + \beta \mathcal{H}(\pi_0(\cdot|s_t))$$
$$(12)$$

where $\mathcal{H}(\pi_0(\cdot|s_t))$ is the policy entropy at state $s_t$. **Thus,** $\mathbb{E}_{a_t \sim \pi_0(\cdot|s_t)}[A_r^{\pi_0}(s_t, a_t)] = -\beta \mathcal{H}(\pi_0(\cdot|s_t))$ **and** $h(\pi_0) = -\mathcal{H}(\pi_0(\cdot|s_t))$**.** The Q function satisfies the Soft Bellman Equation:

$$Q^{\pi_0}(s_t, a_t) = r(s_t, a_t) + \gamma \beta \mathbb{E}_{s_{t+1} \sim P(\cdot|s_t, a_t)} [\mathcal{H}(\pi_0(\cdot|s_{t+1}))]$$
$$+ \gamma \mathbb{E}_{s_{t+1} \sim P(\cdot|s_t, a_t), a_{t+1} \sim \pi_0(\cdot|s_{t+1})} [Q^{\pi_0}(s_{t+1}, a_{t+1})] \quad (13)$$

Denote $f(s_t, a_t) = \gamma \beta \mathbb{E}_{s_{t+1} \sim P(\cdot|s_t, a_t)}[\mathcal{H}(\pi_0(\cdot|s_{t+1}))]$, which can be calculated based on $P$ and $\pi_0$, then we have:

$$Q^{\pi_0} = (I - \gamma P^{\pi_0})^{-1}(r + f) \quad (14)$$

where $Q^{\pi_0}$, $r$, and $f$ are vectors of size $|\mathcal{S}||\mathcal{A}|$, $I$ is an identity matrix, $P^{\pi_0} \in [0,1]^{|\mathcal{S}||\mathcal{A}| \times |\mathcal{S}||\mathcal{A}|}$ is the transition kernel between any two state-action pairs defined with $P$ and $\pi_0$. $I - \gamma P^{\pi_0}$ is invertible, according to Corollary 1.5 in Agarwal et al. [2019]. Combining Eq. (12) and Eq. (14), we have the following linear system:

$$A_r^{\pi_0} + g = (I' - \Pi_0)(I - \gamma P^{\pi})^{-1}(r + f) \quad (15)$$

Here, $A_r^{\pi_0}$, $\log \pi_0$, $r$, $f$ are vectors of $A_r^{\pi_0}(s,a)$, $\log \pi_0(s,a)$, $r(s,a)$, $f(s,a)$, respectively; $I'$ is an identity matrix; $g$ is a vector of size $|\mathcal{S}||\mathcal{A}|$, where each $g(s,a) = \beta \mathcal{H}(\pi_0(\cdot|s))$; $\Pi_0$ is a block diagonal matrix of size $|\mathcal{S}||\mathcal{A}| \times |\mathcal{S}||\mathcal{A}|$, composed of $|\mathcal{S}|$ submatrices, each of size $|\mathcal{A}| \times |\mathcal{A}|$. Specifically, the submatrix corresponding to $s$ is $\Pi_0(s) = [\pi_0(\cdot|s) \cdots \pi_0(\cdot|s)]^T = \mathbf{1}\pi_0(\cdot|s)^T$, where $\mathbf{1}$ is an all-one vector. $I'(s) - \Pi_0(s)$ **is a diagonal block of** $I' - \Pi_0$ **corresponding to state** $s$ **and has a rank of** $|\mathcal{A}| - 1$**.** This is because: (1) $\mathrm{rank}(I'(s) - \Pi_0(s) + \Pi_0(s)) = |\mathcal{A}| \leq \mathrm{rank}(I'(s) - \Pi_0(s)) + \mathrm{rank}(\Pi_0(s)) = \mathrm{rank}(I'(s) - \Pi_0(s)) + 1 \Rightarrow \mathrm{rank}(I'(s) - \Pi_0(s)) \geq |\mathcal{A}| - 1$ and (2) $\mathbf{1}$ is an eigenvector of $I'(s) - \Pi_0(s)$ corresponding to an eigenvalue of 0.

Next, we proof that the linear system $Cx = b$ corresponding to Eq. (15) is consistent, where $C = (I' - \Pi_0)(I - \gamma P^{\pi})^{-1}$, $x = r + f$, and $b = A_r^{\pi_0} + g$. Note that $C$, $f$, and $b$ are defined with $\gamma$, $\beta$, $P$, $\pi_0$, and $\pi$ and so are known. We can apply elementary row operations, represented by a matrix $D$, to the augmented matrix $[C \mid b]$. **In particular,** $D$ **is also a block diagonal matrix, composed of** $|\mathcal{S}|$ **submatrices, each of size** $|\mathcal{A}| \times |\mathcal{A}|$**. The submatrix corresponding to state** $s$ **is an identity matrix with the first row replaced by** $\pi(\cdot|s)^T$**. Notably, the 1st,** $(|\mathcal{A}| + 1)$**-th,** $(2|\mathcal{A}| + 1)$**-th,** $\cdots$**,** $((|\mathcal{S}| - 1)|\mathcal{A}| + 1)$**-th rows of** $D(I' - \Pi_0)$ **(and so** $DC$**) and** $Db$ **are all 0.** This can be easily proved based on the following facts:

$$\pi_0(a|s)(1 - \pi_0(a|s)) - \pi_0(a|s) \left( \sum_{a' \neq a} \pi_0(a'|s) \right) = 0 \Rightarrow \pi_0(\cdot|s)^T(I'(s) - \Pi_0(s)) = \mathbf{0}^T; \quad (16)$$

$$\mathbb{E}_{a \sim \pi_0(\cdot|s)}[A_r^{\pi_0}(s,a)] = -\beta \mathcal{H}(\pi_0(\cdot|s)) \Rightarrow \mathbb{E}_{a \sim \pi_0(\cdot|s)}b(s) = 0.$$

Eliminating these rows in $[DC \mid Db]$, we can get a new augmented matrix $[\tilde{C} \mid \tilde{b}]$ where $\tilde{C}$ has a full row rank (based on the facts that $\mathrm{rank}(I'(s) - \Pi_0(s)) = |\mathcal{A}| - 1$, $\forall s$ and $I - \gamma P^{\pi}$ is invertible). Thus, the original linear system $Cx = b$ (i.e., Eq. (15)) is consistent, according to the Rouché-Capelli theorem. □

## C  Related Works

Table 1: We propose RICL for LLMs to learn from environmental feedback, fundamentally different from intrinsic self-correction work like SELF-REFINE Madaan et al. [2024]. Unlike STaR [Zelikman et al., 2022] and Reflexion [Shinn et al., 2024], our method utilizes environmental feedback to retrospectively update the policy. Additionally, we do not fine-tune an extra reflector, as done in RLMEC [Chen et al., 2024] and Retroformer [Yao et al., 2023]. By generating dense training signals from sparse environmental feedback, our approach is more sample-efficient than methods that rely solely on sparse preference rewards, such as Iterative RPO [Pang et al., 2024].

|  | Involving Env Feedback | Requiring an Extra Reflector | Retrospective Updating | Multi-turn Credit Assignment |
|---|---|---|---|---|
| SELF-REFINE | ✗ | ✗ | ✗ | ✗ |
| STaR, Reflexion | ✓ | ✗ | ✗ | ✗ |
| RLMEC, Retroformer | ✓ | ✓ | ✗ | ✗ |
| Iterative RPO, RICO-GRPO | ✓ | ✗ | ✓ | ✗ |
| RICL (Ours) | ✓ | ✗ | ✓ | ✓ |

## D  Comparison of Training Times

Table 2: Training time (in minutes) required for each algorithm on different tasks, along with the GPU setup used.

| Method | goto | pickup | pick_up_seq_go_to | open | GPU Type |
|---|---|---|---|---|---|
| RWR | 72 | 69 | 232 | 227 | NVIDIA A40 × 2 |
| RICOL | 96 | 95 | 594 | 585 | NVIDIA A40 × 2 |
| PPO (10M) | 258 | 257 | 234 | 198 | GeForce RTX 2080 Ti × 1 |
| PPO (3B) | 1440 | 1440 | 1440 | 1440 | NVIDIA A40 × 4 |

## E  Environments

### E.1  1D Key-Door Environment

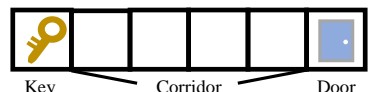

Figure 7: An illustration for the 1D Key-Door scenario.

As illustrated in Figure 7, the environment consists of a 1D grid world where the agent can traverse along the x-axis. The key is located at the leftmost grid cell, and the door is positioned at the rightmost grid cell. Initially, the agent starts without holding the key, aiming to move left to retrieve the key and subsequently move right to unlock the door. The corridor length used in the paper is $10$. The available action includes {move left, move right, pick up the key, unlock the door}. The prompt can be found in Figure 8 and Figure 9.

For any given policy, we can compute the corresponding transition matrix $P$, where the entry in the $i$-th row and $j$-th column represents the probability of the agent transitioning from state $s_i$ to state $s_j$ in a single step. The ground truth value function $V$ can then be computed as: $V = (I - \gamma P)^{-1} R$, where $\gamma$ is the discount factor and $R$ is the reward function. Once the value function is determined, and given that the environment is deterministic, we can compute the advantage function using the following formula: $A(s, a) = V(s) - r(s, a) - \gamma V(s')$ where $s'$ is the next state resulting from applying action $a$ in state $s$, and $r(s, a)$ is the immediate reward received when transitioning from $s$ to $s'$ by taking action $a$.

You are a helpful navigation agent.
Please thoroughly review your goal, the available actions, objects around you, and any advice from experts, if available, and respond to the question strictly adhering to this specified format: ACTION: [your_answer]

Your goal is to open the locked door. You can choose to: **move left, move right, pick up the key, or unlock the door**. You see the locked door 200 meters away on your right. You don't have the key. You see a key on the ground around you. After reviewing the rules of the environment, your goal and objects around you, what do you do next to achieve the goal?

Figure 8: The prompt used for the LLM policy in the 1D Key-Door scenario.

I will provide you with a trajectory starting from step 0 and the state at step 0.
Firstly, you have to analyze the task, the goal, and the current state, generate a rough plan.
Then, based on this trajectory, you need to evaluate whether the decision you made at step 0 was correct.
Additionally, if given another opportunity, how would you modify it, if needed?
Provide concise verbal feedback as a guideline for others new to this task and facing the same state in the future.
If the policy is generally correct, you can reply with "No modifications needed."
After providing your explanation, output your final feedback by strictly following this format: "Verbal Feedback: [your_answer]"

Trajectory/state (skip due to the space limit)

Figure 9: The prompt used for the LLM reflector in the 1D Key-Door scenario.

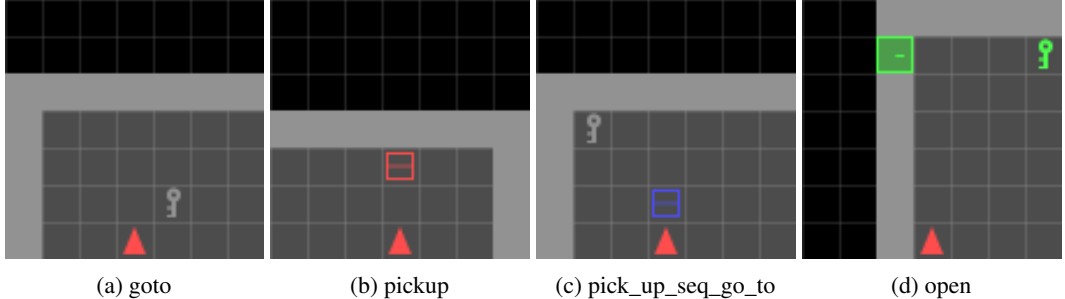

|(a) goto|(b) pickup|(c) pick_up_seq_go_to|(d) open|

Figure 10: Screenshots of four BabyAI scenarios, where the agent is partial-observable.

## E.2   BabyAI Environment

We evaluate our algorithm in the BabyAI environment [Chevalier-Boisvert et al., 2018], a 2D grid world where the player navigates an agent to accomplish specified tasks. We use BALROG's [Paglieri et al., 2024] implementation of the environment. We test our algorithm on four scenarios: *goto*, *pickup*, *pick_up_seq_go_to*, and *open*. As shown in Figure 10, the player directs the agent (a red triangle) toward a specified local object or manipulate it. The environment is partially observable, with the agent having a 7x7 window of visibility in front of it, viewed from an egocentric perspective. The agent can perform six actions: Turn Left, Turn Right, Move Forward, Pickup, Drop, and Toggle. Both input and output for the tasks are presented in text form. The prompt can be found in Figure 11 and Figure 12. We did two editions to the BabyAI environment. Firstly, following Ren et al. [2023], we modify the policy prompt so that its output is constrained to a single character between A and F, with each character corresponding to a specific action. This ensures that the probability of each action is not biased by differences in token length. Secondly, we set the maximum episode length to 16 for the *goto* and *pickup* tasks, and 32 for the *pick_up_seq_go_to* and *open* tasks. This early truncation improves sample efficiency. All baseline algorithms are evaluated using the updated environment to ensure a fair comparison.

---

System Prompt

You are an agent playing a simple navigation game. Your goal is to {mission}.
The following are the possible actions you can take in the game, followed by a short description of each action:
"A": "turn to the left",
"B": "turn to the right",
"C": "take one step forward",
"D": "pick up the object one step in front of you",
"E": "drop the object that you are holding",
"F": "manipulate the object in front of you",
In a moment I will present you an observation. Tips:
- Once the desired object you want to interact or pickup in front of you, you can use the 'toggle' action to interact with it.
- It doesn't make sense to repeat the same action over and over if the observation doesn't change.
- answer the alphanumerical action, not the description.

PLAY!

User Prompt

Current Observation:
a wall 3 steps forward
a wall 1 step right

You always have to output one of the above actions at a time and no other text. You always have to output an action until the episode terminates.

Figure 11: The prompt used for the LLM policy in all the BabyAI scenarios.

You are an agent playing a simple navigation game. Your goal is to {mission}.
The following are the possible actions you can take in the game, followed by a short description of each action:
"A": "turn to the left",
"B": "turn to the right",
"C": "take one step forward",
"D": "pick up the object one step in front of you",
"E": "drop the object that you are holding",
"F": "manipulate the object in front of you",

You are a coach that can provide feedback to the agent.
I will provide you the observation at time step t and the action taken by the agent at time step t.
Then I will provide you the trajectory after that.
Your task is to carefully analyze the trajectory and reflect on the agent's decision at time step t.
If the agent were to face the same situation again (observation at time step t), what advice would you give to better achieve the goal?
Provide your feedback as a concise and informative string.

The observation at time step t is: {o_t}
The action taken by the agent at time step t is: {a_t}
The trajectory after that is: {traj}

Provide a concise, specific verbal feedback that can help the agent improve its performance when it encounters state t again to help it achieve the goal.
Breifly analyze the agent's decision at time step t and the consequences of the action then provide the feedback.

You should always follow the format:
Your goal: [your goal]
state t: [the state at time step t]
action t: [the action taken by the agent at time step t]
state t+1: [the state at time step t+1]
analyze: [The agent took action t in state t and transitioned to state t+1. Analyze the decision and its consequences, does the action help the agent achieve the goal or make progress?]
Conclusion: [Based on the helpfulness, will you suggest the agent to take action t again, why?]
Feedback: [your concise verbal feedback that can help the agent improve its performance]

Figure 12: The prompt used for the LLM reflector in all the BabyAI scenarios. Only the **Feedback** part is extracted and used for RICL's in-context policy updates.

## F    PPO Implementation Details

We implement PPO (3B) based on the VeRL RLVR framework [Sheng et al., 2025], integrated with our own multi-turn dialogue pipeline. Below, we outline the key implementation details.

**State:** We largely follow the prompt template provided by the BALROG benchmark [Paglieri et al., 2024]. To efficiently manage context length, we include only the two most recent state–action pairs in the prompt. RICOL adopts the same context management strategy to ensure a fair comparison between our method and the baseline.

**Reward:** We employ a sparse binary reward scheme, assigning a single positive reward at the end of a successful trajectory and zero otherwise. This reward structure is applied consistently to both our method and the baselines. Unlike PPO, our algorithm additionally leverages dense implicit rewards derived from the LLM policy, improving sample efficiency.

**Actions:**   The environment features a discrete action space consisting of `move_forward`, `turn_right`, `turn_left`, `pickup`, `drop`, and `toggle`. We assign a penalty of $-0.1$ to LLM agents when they select an invalid action.

## G    RICL can Identify Critical States in Sequential Decision-Making

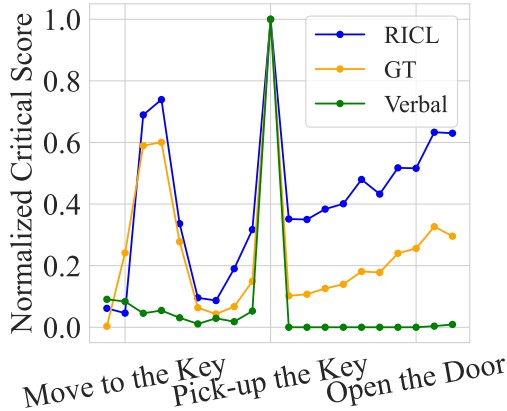

Figure 13: Comparison of critical states identified by RICL and verbal credit assignment. The x-axis represents a sequence of states: the agent first moves toward the key over 9 steps, picks up the key, then moves toward the door over another 9 steps, and finally opens the door. The y-axis indicates the score reflecting how critical each state is, where GT denotes the ground truth. The results show that both algorithms correctly identify the "pick up the key" action as a critical state-action pair. However, only RICL additionally identifies some of the earlier states in the "move to the key" phase as critical.

Here, we demonstrate that RICL assigns higher credits to critical states in the environment. Critical states are defined as those where policy adjustments lead to greater performance gains. For the sampling policy $\pi_0$ and an improved policy $\pi$, $D_{KL}(\pi(\cdot|s)||\pi_0(\cdot|s))$ serves as a measure of how significantly the policy needs to be adjusted at state $s$ to achieve policy improvement (from $\pi_0(\cdot|s)$ to $\pi(\cdot|s)$). Thus, it can be used as a criterion for identifying critical states. Specifically, $\pi$ can be $\pi_{gt}$ or $\pi_{RICL}$ (as defined in Section 6.2) to identify the critical states with the ground-truth policy or the RICL-updated policy, respectively.

Figure 13 depicts the (normalized) critical score (i.e., $v^\pi = D_{KL}(\pi(\cdot|s)||\pi_0(\cdot|s))$) in the 1D Key-Door scenario using LLaMA-3.1-8B-Instruct as $\pi_0$. The yellow curve shows two peaks in the critical score $v^{\pi_{gt}}$. The right peak corresponds to the state – picking up the key, which is critical in the 1D Key-Door scenario. The left peak corresponds to a state where the expert decides to move toward the key. Due to imperfections in $\pi_0$, this state also becomes critical. The blue curve in Figure 2 represents $v^{\pi_{RICL}}$. The peaks of the blue curve align with those of the yellow curve (i.e., the ground truth), demonstrating that RICL effectively identifies critical states.

We further compare RICL with methods that explicitly utilize LLMs for credit assignment [Pignatelli et al., 2024] (represented by the green curve in Figure 13). Specifically, these methods prompt an LLM with a trajectory and ask it to identify the critical state within the sequence. We evaluate this approach on 1000 distinct trajectories, using the frequency with which a state $s$ is labeled as critical to compute its critical score. As shown by the green curve in Figure 13, this method can only identify the "pick-up-the-key" state as critical. It fails to detect the left peak of the yellow curve. This limitation arises because identifying this critical state requires knowledge of the sampling policy $\pi_0$, which is provided to but overlooked by this baseline approach.

## H   Comparisons between Retroformer and RICOL

First, Retroformer [Yao et al., 2023] fine-tunes a reflector LLM to generate prompts for a fixed actor LLM, whereas our method fine-tunes an actor LLM using guidance from a fixed reflector LLM. The objectives of the two approaches are fundamentally orthogonal. Second, Retroformer relies on trajectory-level sparse rewards for LLM fine-tuning, whereas RICL leverages step-level dense training signals. Specifically, Retroformer requires rolling out two trajectories – one with reflector-generated hints ($\tau_1$) and one without ($\tau_2$) – and computes the reward as the return difference between the two, which is then used to fine-tune the reflector. In contrast, our method applies step-level dense supervision by estimating the advantage function at each time step. This is computed based on the change in policy before and after the in-context update: $\log \frac{\pi_{\text{in-context updated}}(a|s)}{\pi_{\text{sampling}}(a|s)}$. **In short, for an episode of length $n$, RICL collects $n$ supervised signals from $n$ environment steps, while Retroformer requires $2n$ environment steps to obtain just one supervision signal.**

Retroformer estimates $\Delta(s_0, \text{feedback}) = V^{\pi_{\text{in-context updated}}}(s_0) - V^{\pi_{\text{sampling}}}(s_0)$, i.e., the value difference between two policies, using the difference in trajectory returns: $Reward(\tau_1) - Reward(\tau_2)$. In contrast, our method only requires estimating the advantage function under a single policy, $\pi_{\text{sampling}}$, defined as $A(s, a) = Q^{\pi_{\text{sampling}}}(s, a) - V^{\pi_{\text{sampling}}}(s)$. This is generally more sample-efficient. The following experiments use the Monte Carlo method to estimate both quantities. The table below reports the mean squared error (MSE) between the estimated and ground truth values of $\Delta(s_0, \text{feedback})$ and $A(s, a)$.

Table 3: Approximation errors vs. number of sample trajectories.

| Number of Sample Trajectories | 1,000 | 10,000 | 100,000 | 1,000,000 |
|---|---|---|---|---|
| MSE on $A(s, a)$ | 0.1287 | 0.0492 | 0.0492 | 0.0454 |
| MSE on $\Delta(s_0, \text{feedback})$ | 0.2763 | 0.2788 | 0.2713 | 0.2641 |

The results indicate that, under Monte Carlo sampling, Retroformer requires significantly more trajectories than our method to achieve accurate value estimations. Further, as shown in Figure 2 of the main paper, our actual sampling strategy (i.e., RICL) achieves over $100\times$ greater sample efficiency compared to the Monte Carlo baseline. Altogether, these findings demonstarte that our approach is substantially more sample-efficient than Retroformer in terms of value estimation.

