# OpenReview forum: "Retrospective In-Context Learning for Temporal Credit Assignment with Large Language Models"
_NeurIPS.cc/2025/Conference — NeurIPS 2025 poster_

### Official Review · Reviewer_nKQa · 2025-06-06

**Clarity:** 3
**Significance:** 2
**Originality:** 2
**Rating:** 3
**Confidence:** 2

**Summary:**

Thus uses a pretrained LLM to read a hindsight trajectory (future states, actions, rewards) and generate textual feedback, turning sparse, delayed rewards into dense advantage estimates without a separate value network. Building on this, “Retrospective In-Context Online Learning (RICOL)” alternates between data collection, RICL-based advantage estimation, and advantage-weighted policy updates. Empirically, RICL matches ground-truth advantages with ∼10 trajectories (∼100× more sample-efficient than Monte Carlo) in a toy task, and RICOL outperforms reflection-based and PPO baselines on BabyAI benchmarks, remaining robust even when the feedback is noisy.

**Questions:**

- I’m concerned about the consistency between the RL motivation and the algorithm. Section 4 treats the ICL‐based update as a one‐step soft policy update, but since the implicit reward is generated by the updated policy itself, the reward changes each iteration. In other words, ICL does not genuinely perform maximum‐entropy RL because the reward is not well‐defined. Can the authors clarify this?
- Substituting (3) into (1) gives $\exp(A)\approx\frac{\pi'}{\pi_0}$. To recover $\pi'$ on the RHS, shouldn’t the update be $\pi_{k+1}\propto \pi_k\,\exp(A)$, which is the closed‐form for KL‐regularized policy updates in standard RL rather than maximum‐entropy RL? Am I misunderstanding something?
- Computing $Z(s)$ at every iteration seems costly. Is this necessary, or can it be avoided without imposing a significant implementation burden?

**Ethical Concerns:**

["NO or VERY MINOR ethics concerns only"]

**Limitations:**

I would suggest the author to provide more detailed experimental setups, e.g., how the reward is chosen for PPO training in Section 6.4.

**Paper Formatting Concerns:**

/

**Quality:**

3

**Strengths And Weaknesses:**

- This paper introduce a novel framework RICL motived by the RL by understanding the improvement by the ICL context fromth view of implicit reward guidance from the view of one-step soft policy iteration.
- The paper contains comprehensive comparison on BabyAI benchmark, which validates the superiority of the proposed method.
- The RL motivation is not consistent with the proposed method, and I will defer the clarifying questions into later part.

---

> ### Author Rebuttal · Authors · 2025-07-31
>
> We sincerely thank the reviewer for their time and thoughtful evaluation of our work. We also appreciate the positive feedback on our empirical results, particularly the strong performance demonstrated on the BabyAI benchmark. Below, we provide detailed responses to further clarify the motivation and technical details behind our approach.
>
> > Q1: The implicit reward is generated by the updated policy itself and changes every iteration. Doesn't this mean the reward is not well-defined, and thus ICL does not genuinely perform maximum-entropy RL?
>
> - First, the ground-truth environment reward is used during RL training. However, in benchmarks like BabyAI, these rewards are sparse, resulting in high sample complexity. To address this, we introduce implicit rewards to improve credit assignment and sample efficiency.
> - Prior work on RL for LLMs commonly uses implicit rewards via Process Reward Models (PRMs). For example, [1, 2] define the reward as $r_{\phi}(y_t) := \beta \log \frac{\pi_{\phi}(y_t | y_{<t})}{\pi_{\text{ref}}(y_t | y_{<t})},$ where $\pi\_{\phi}$ is the current policy and $\pi\_{\text{ref}}$ is a reference policy. This reward evolves with the policy during training, as ours.
> - More broadly, implicit rewards in RL are often used to encourage the policy to explore regions that it has previously less visited, making the implicit rewards inherently policy-dependent [3]. Therefore, our implicit reward design, which depends on the policy itself, is well-defined.
>
>
> > Q2: Substituting (3) into (1) gives exp(A) ≈ π′ / π₀. To recover π′, shouldn’t the update be πₖ₊₁ ∝ πₖ · exp(A), which matches the closed-form for KL-regularized updates in standard RL? Doesn’t this suggest the method aligns with KL-regularized RL rather than maximum-entropy RL?
>
> - The reviewer is correct: this corresponds to KL-regularized policy updates, not maximum-entropy RL. We will fix this misstatement in the next revision of the paper.
> - Importantly, this misstatement does not affect the correctness of the proposed algorithm or validity of the empirical results.
> - We originally referenced maximum-entropy RL (MERL) in the main paper as an example of closed-form policy improvement, but we acknowledge that the connection between the preliminary discussion on MERL and our proposed method may not be immediately clear. To avoid confusion, we will move this discussion to the appendix in the next version.
>
> > Q3: The RL motivation is not consistent with the proposed method.
>
> The design of RICOL is fundamentally motivated by two core concepts in reinforcement learning: credit assignment and online learning (for policy improvement).
>
> - Retrospective In-Context Learning for Credit Assignment: RICOL uses RICL to infer the advantage function at each iteration. This mechanism enables credit assignment at each time step, which significantly improves sample efficiency in environments with sparse rewards.
>
> - Online Policy Updates via Advantage-Weighted Regression (AWR): We perform policy improvement using the closed-form solution of KL-regularized updates, where the update is weighted by the inferred advantage function computed by RICL.
>
> Credit assignment and online learning are fundamental aspects that distinguish RL from supervised learning. RICOL is explicitly designed to address both challenges. Our results on the BabyAI benchmark, where we achieve state-of-the-art performance, demonstrate that this RL-inspired approach can effectively learn from sparse external rewards.
>
> > Q4: Computing $Z(s)$ at every iteration seems costly. Is this necessary, or can it be avoided without imposing a significant implementation burden?
>
> - Computing $Z(s)$ is necessary. This is because evaluating the log-probability of each token requires knowledge of the normalization factor $Z(s)$.
> - Recent RLHF work [4] also requires computing $Z(s)$ in their algorithm design.
> - The associated computational overhead for computing $Z(s)$ remains tractable on modern hardware. To demonstrate this, we have reported the total wall-clock training time (in minutes) in Appendix D.
> | Method     | goto | pickup | pick_up_seq_go_to | open | GPU Type                   |
> |------------|------|--------|--------------------|------|----------------------------|
> | RWR        | 72   | 69     | 232                | 227  | NVIDIA A40 × 2            |
> | RICOL      | 96   | 95     | 594                | 585  | NVIDIA A40 × 2            |
> | PPO (3B)   | 1440 | 1440   | 1440         | 1440 | NVIDIA A40 × 4            |
>
>
> > Q5: More detailed experimental setups for PPO training in Section 6.4.
>
> We implemented PPO (3B) based on VeRL’s RLVR framework [5], with our own multi-turn dialogue pipeline. Below, we elaborate on key implementation details, which will be included in the next version of the paper:
> - State: We largely reuse the prompt template provided by the BALROG benchmark [6]. To manage context length efficiently, we include only the two most recent state-action pairs in the prompt, following a strategy similar to that used in GiGPO [7]. RICOL also adopts this context management approach.
> - Reward: We use a sparse binary reward signal, assigning a single positive reward at the end of a successful trajectory and zero otherwise. This reward scheme is applied to both our algorithm and the baselines. Unlike PPO, our algorithm can leverage additional dense implicit rewards derived from the LLM policy, leading to improved sample efficiency.
>
>
> [1] Yuan, Lifan, et al. "Free process rewards without process labels." arXiv preprint arXiv:2412.01981 (2024).
>
> [2] Cui, Ganqu, et al. "Process reinforcement through implicit rewards." arXiv preprint arXiv:2502.01456 (2025).
>
> [3] Burda, Yuri, et al. "Exploration by random network distillation." arXiv preprint arXiv:1810.12894 (2018).
>
> [4] Ouyang, Long, et al. "Training language models to follow instructions with human feedback." Advances in neural information processing systems 35 (2022): 27730-27744
>
> [5] Sheng, Guangming, et al. "Hybridflow: A flexible and efficient rlhf framework." Proceedings of the Twentieth European Conference on Computer Systems. 2025.
>
> [6] Paglieri, Davide, et al. "Balrog: Benchmarking agentic llm and vlm reasoning on games." arXiv preprint arXiv:2411.13543(2024).
>
> [7] Feng, Lang, et al. "Group-in-group policy optimization for llm agent training." arXiv preprint arXiv:2505.10978 (2025).

---

> > ### Comment · Area_Chair_bX1p · 2025-08-04
> > **Please respond to the author's rebuttal post**
> >
> > Hi Reviewer nKQa, I see no response letting me know whether or not the rebuttal has changed your opinion. Could you please let me and the authors know by engaging? This process is critical to enabling the (S)ACs to make a decision on this work.
> >
> > --Your AC

---

> ### Author Response · Authors · 2025-08-05
>
> Thank you for your feedback and for recognizing the empirical improvements of our method. Could you kindly clarify which part of the theoretical motivation you find unconvincing? We would be happy to provide further explanation or clarification.

---

### Official Review · Reviewer_Vawv · 2025-07-01

**Clarity:** 3
**Significance:** 3
**Originality:** 3
**Rating:** 4
**Confidence:** 3

**Summary:**

This paper investigates how to utilize the in-context learning ability of LLMs to do credit assignment in a more informative way when learning under sparse reward. It first updates per-step advatange function by utilizing the policy change before and after in-context learning, then updates the policy via AWR algorithm. Empirical results on four BabyAI tasks show that this method outperforms baseline methods w.r.t. both sample efficiency and final performance on most tasks.

**Questions:**

1. Theorem 4.1 proves the existence of a reward function that is proportional to the log probability change of the policy before and after in-context learning. But it does not mean that this reward function will be the same as the true reward function of the MDP right? Empirically they seem to align well, but logically how do you ensure this?
2. The proposed method is more sample efficient than the baseline methods. But I was wondering that how do they compare w.r.t. the total computational cost? As your method needs to reflect on each step of each collected episode, I'll expect that it will consume more tokens than the baseline methods, which either reflect on the whole trajectory once or do no reflection right? As you are evaluating on language games in simulation, the total computational cost may be a more important metric to reduce compared to the sample cost, as they are just tokens and have no difference practically.
3. In Section 6.3, how do you drawer the conclusion that RICL is safer than ICL based on its higher accuracy? I agree that it's more effective, but I'm not clear about the logic of how you can say that it's also safer based on a single accuracy score.
4. In figure 4, there seems to be no result for PPO (3B) on the "open" task?

**Ethical Concerns:**

["NO or VERY MINOR ethics concerns only"]

**Final Justification:**

I have gone through the rebuttal from the authors and am satisfied with it in general. I did not raise my score to 5 mainly because that there is no empirical results showing that if the methodology can be helpful in more valuable domains other than multi-turn language games. But I think the methodology proposed by the paper is well-motivated, novel and has good potential, and will highly recommand acceptance of the paper.

**Limitations:**

Yes

**Quality:**

3

**Strengths And Weaknesses:**

Strengths:
1. The paper is well-motivated and proposes a novel way to enable temporal credit assignment with in-context learning, which is well supported by theoretical derivation.
2. The experiments align well with the methodology design, which validate the importance of the key design choices in the algorithm, and the improvement over the baselines is generally significant and well supports the motivation of the paper.

Weaknesses:
1. The evaluation tasks considered in the paper are relatively limited, as also discussed by the authors in the conclusion section. The effectiveness of the algorithm heavily depends on the in-context learning ability of the LLMs, so I think it's important to see if such ability can scale up to and benefit more complicated tasks like math reasoning.

---

> ### Author Rebuttal · Authors · 2025-07-31
>
> We sincerely thank the reviewer for the time and effort spent in reviewing this manuscript. We also appreciate the reviewer’s recognition of the paper’s strengths, including the well-motivated approach, solid theoretical support, and the strong empirical validation of key design choices. Below, we provide detailed responses to the reviewer’s concerns:
>
> > Q1: The evaluation tasks considered in the paper are relatively limited, as also discussed by the authors in the conclusion section. The effectiveness of the algorithm heavily depends on the in-context learning ability of the LLMs, so I think it's important to see if such ability can scale up to and benefit more complicated tasks like math reasoning.
>
> RICOL is designed to address multi-turn credit assignment problems. However, there are currently no off-the-shelf math reasoning benchmarks that require multiple dialogue turns between the environment and LLM agents.
>
> We added an additional experiment on two scenarios from the Baba Is AI benchmark [1], a grid-based puzzle environment inspired by a video game, Baba Is You. This benchmark is designed to test agents' reasoning and planning abilities in dynamic environments where rules can be rearranged during an episode. Each episode lasts 100 steps, and agents must adapt with dynamic environments by combining previously seen rules in novel ways—making it a challenging testbed for evaluating reasoning capabilities of LLM agents on long horizon tasks.
>
> Scenario: `goto_win`
> | Algo Name    | Max Succ. Rate (%) | First Step That Achieves Max Succ. Rate | Succ. Rate at Time Step 7680 (%) |
> | ------------ | -------------- | --------------------------------------- | ---------------------------- |
> | RICOL (ours) | 100.0 ± 0.0        | 7680                                    | 100.0 ± 0.0                      |
> | PPO (3B)     | 100.0 ± 0.0        | 87040                                   | 43.8 ± 4.4                   |
>
> Scenario: `goto_win-distr_obj`
> | Algo Name    | Max Succ. Rate (%) | First Step That Achieves Max Succ. Rate | Succ. Rate at Time Step 14848 (%) |
> | ------------ | -------------- | --------------------------------------- | ---------------------------- |
> | RICOL (ours) | 95.3 ± 1.9     | 14848                                   | 95.3 ± 1.9                   |
> | PPO (3B)     | 100.0 ± 0.0        | 107520                                  | 28.9 ± 2.0                   |
>
> Results show that RICOL achieves comparable performance to PPO (3B) on these scenarios while being ~10 times more sample efficient, highlighting its effectiveness on more complex, reasoning-intensive tasks.
>
> > Q2: Theorem 4.1 proves the existence of a reward function that is proportional to the log probability change of the policy before and after in-context learning. But it does not mean that this reward function will be the same as the true reward function of the MDP right? Empirically they seem to align well, but logically how do you ensure this?
>
> - In the policy improvement step of RICOL, we use the output of the in-context updated policy as an estimate of the advantage function (Eq. 3) to guide the training of the actor policy. This approach assumes the existence of a reward function whose corresponding advantage function explains the discrepancy between the in-context updated policy and the actor policy. Theorem 4.1 guarantees the existence of such a reward function for ANY pair of policies, thereby supporting the soundness of our algorithm's design.
> - Our policy improvement step (Eq. 5) closely resembles advantage-weighted regression and thus relies solely on the accurate estimation of the advantage function. Empirically, we show that RICL yields a close approximation of the true advantage function in section 6.2.
> - A single advantage function may correspond to multiple reward functions [2]. Fortunately, our method only requires consistency at the advantage level and does not rely on recovering the true underlying reward function. Thus, we do not need to provide guarantee on the reward consistency.
>
>
>
> > Q3: The proposed method is more sample efficient than the baseline methods. But I was wondering that how do they compare w.r.t. the total computational cost? As your method needs to reflect on each step of each collected episode, I'll expect that it will consume more tokens than the baseline methods, which either reflect on the whole trajectory once or do no reflection right? As you are evaluating on language games in simulation, the total computational cost may be a more important metric to reduce compared to the sample cost, as they are just tokens and have no difference practically.
>
> For the tasks we evaluate on, the simulation cost is negligible, and the main computational bottleneck lies in the inference time of the LLMs. To provide a fair comparison, we report the total wall-clock training time (in minutes) in Appendix D.
>
> | Method     | goto | pickup | pick_up_seq_go_to | open | GPU Type                   |
> |------------|------|--------|--------------------|------|----------------------------|
> | RWR        | 72   | 69     | 232                | 227  | NVIDIA A40 × 2            |
> | RICOL      | 96   | 95     | 594                | 585  | NVIDIA A40 × 2            |
> | PPO (3B)   | 1440 | 1440   | 1440         | 1440 | NVIDIA A40 × 4            |
> | PPO (10M)  | 258  | 257    | 234            | 198  | GeForce RTX 2080 Ti × 1   |
>
>
> Our results show that RICOL requires approximately 1.5× to 2× more training time than the RWR baseline. This overhead primarily stems from the step-wise reflection process. However, RICOL is significantly more efficient than PPO (3B), as it converges in far fewer environment steps, thereby saving substantial policy inference time, offsetting the reflection cost. We also want to emphasize that in many real-world scenarios, such as robotics or scientific discovery, simulation or data collection can be expensive. In such cases, sample efficiency (i.e., requiring fewer environment steps) is often a more critical metric than token efficiency. By enabling faster policy adaptation with fewer samples, our method offers practical benefits in these domains.
>
> > Q4: In Section 6.3, how do you draw the conclusion that RICL is safer than ICL based on its higher accuracy? I agree that it's more effective, but I'm not clear about the logic of how you can say that it's also safer based on a single accuracy score.
>
> - We agree that the term “safer” may be misleading in this context and will revise it to “more reliable” in the next version of the paper.
> - In Section 6.3 of the main paper, we compare the in-context updated policies produced by RICL and standard ICL, and show that RICL produces a more reliable policy—one that is closer to the optimal policy—on the BabyAI task. The accuracy score quantifies the distance between the updated and optimal policies.
> - Our intention was to express caution regarding LLM-based in-context learning, which is often prone to hallucinations, particularly on previously unseen states. RICL addresses this issue through a retrospective update strategy that updates the policy only at the state just encountered. In contrast, standard ICL implicitly assumes generalization across unseen states. In this sense, RICL offers a more reliable update mechanism for LLM agents.
>
> > Q5: In figure 4, there seems to be no result for PPO (3B) on the "open" task?
>
> PPO (3B) fails to learn effectively on the open task due to the extremely low zero-shot success rate of the underlying policy, i.e., Llama-3.2-3B-Instruct, and the sparsity of rewards. As a result, its learning curve remains flat and largely overlaps with those of other tasks, making it visually indistinguishable in Figure 4. We will clarify this in the revised version of the paper.
>
> [1] Cloos, Nathan, et al. "Baba is AI: Break the rules to beat the benchmark." arXiv preprint arXiv:2407.13729 (2024).
>
> [2] Ng, Andrew Y., Daishi Harada, and Stuart Russell. "Policy invariance under reward transformations: Theory and application to reward shaping." Icml. Vol. 99. 1999.

---

> ### Comment · Reviewer_Vawv · 2025-08-04
>
> Dear authors,
>
> Thanks for your detailed reply and extended experiments!
>
> I was wondering that if the current version of your method can be applied to any other domain other than multi-turn language games? For example, can your method be used to improve the CoT when solving math problems? Please correct me if I'm wrong, but based on my understand, your method is limited to tasks where the action in each step (turn) can be expressed as a verb right? So it is hard to say what is the action in a CoT step where the policy (LLM) outputs a sequence of tokens?
>
> My main concern about the submission is to what domains is the method applicable. So if the authors can help address the above questions, I'm happy to further raise my score given the greater value of the method.

---

> ### Author Response · Authors · 2025-08-04
>
> Thank you for your insightful questions and constructive feedback.
> Our method, RICOL, is designed for multi-turn language games by (1) generating trajectories via environment interaction, (2) producing verbal feedback for each state-action pair using a Reflector model and hindsight trajectories, (3) in-context update the policy $\pi’_{k+1}$ using this feedback, and (4) distilling it into the model parameters via an AWR-style loss, as shown in Eq. (5).
>
> While our experiments focus on language games, RICOL can, in principle, be adapted to other domains such as math reasoning with chain-of-thought (CoT). The key differences are: (a) Transition from an MDP described in natural language to a token-level MDP, where the action corresponds to the next token and the state is the previously generated sequence; (b) generating feedback for every token is computationally intensive and less interpretable. To address this, one could combine the Reflector with a Verifier to identify critical reasoning steps in the CoT trace and provide targeted feedback only on those.
>
> To sum up, we only need to modify step (2) in RICOL while keeping the rest intact to adapt to the use for math reasoning. Applying RICOL to such tasks is promising, though further prompt/verifier design would be needed, which we leave as future work.

---

> > ### Comment · Reviewer_Vawv · 2025-08-05
> >
> > Dear authors,
> >
> > Thanks for your detailed reply and explanation!
> >
> > Applying your method to math reasoning as proposed in your reply sounds promising. But I'm not sure how well it can work practically given that token-level action is much less interpretable and more expensive computationally (as you've also mentioned), and it seems not easy to determine what is a reasoning step, verify the correctness of it, and correct it in an efficient and high-quality way. As there is no empirical result supporting the applicability of the method to more domains yet, I'll keep my score but highly recommand acceptance of the paper given the high potential and novelty of the method.

---

> > > ### Author Response · Authors · 2025-08-05
> > >
> > > Thank you for your thoughtful feedback. We appreciate your recognition of our method's potential and novelty, and we see extending it to broader domains as a promising direction for future work.

---

### Official Review · Reviewer_b3Y6 · 2025-07-02

**Clarity:** 3
**Significance:** 3
**Originality:** 3
**Rating:** 5
**Confidence:** 2

**Summary:**

This paper proposes a novel method called retrospective in-context learning (RICL) to address the temporal credit assignment problem in reinforcement learning with large language models (LLMs). It effectively tackles language-conditioned, sparse-reward decision-making tasks with high sample efficiency. The proposed method first transforms sparse reward signals into advantage functions and then leverages these advantage functions for weight regression to extract additional informative signals. Empirical results demonstrate that RICL achieves both efficacy and efficiency in solving decision-making tasks.

**Questions:**

1. Can you elaborate more on the computational overhead of the RICL? Will it take much longer inference time than the other methods mentioned in the paper?
2. Can you elaborate more on the difference between the RICL and the previous methods mentioned in the paper, such as the Reflexion and RICO-GRPO?
3. Can you provide the PPO(7B) model results to have a fairer comparison with the RICL in BabyAI?
4. Can you explain why the open task performs worse than the GPT-4o-mini while the other three tasks perform better than the GPT-4o-mini in Figure 4? Is there any specific reason for this observation?

**Ethical Concerns:**

["NO or VERY MINOR ethics concerns only"]

**Final Justification:**

The authors’ rebuttal and added experiments address my main concerns. (1) On complex environments, RICOL outperforms RWR and PPO in sample efficiency and convergence, and wall-clock time is now reported. (2) Clear clarification on the difference with existing methods solved my concerns.

**Limitations:**

yes

**Paper Formatting Concerns:**

No formatting concerns

**Quality:**

3

**Strengths And Weaknesses:**

**Strengths:**

1. This paper demonstrates that the ratio of two policies can be used to estimate the advantage function and the advantage function can be used to solve the temporal credit assignment problem.
2. The empirical results show that RIOCL outperforms the baselines in the BabyAI scenarios.  Experiments on comparing RICL with the base model and the ICL method show the effectiveness of the RICL. Compared with the MC method, RICL achieves an accurate advantage estimation with fewer samples.
3. The paper is well-structured and clear enough.
4. The paper provides a novel method to solve the temporal credit assignment problem in sparse-reward decision-making tasks with high sample efficiency. It can provide a new technique for LLM agents to solve the sparse-reward decision-making tasks.
5. The idea of merging Retrospective In-Context Learning (RICL) with Advantage-Weighted Regression to form a full online RL loop is novel and interesting.


**Weaknesses:**

1. The empirical study demonstrates effectiveness in structured environments such as BabyAI but lacks evaluation in more complex, real-world scenarios.

2. The proposed methods need an additional reflector LLM and a second forward pass of the policy, which may introduce significant computational overhead.

3. The paper compares the RIOCL with the PPO(3B) in Figure 4, which seems to be unfair. The base model should have the same size as the RIOCL.

---

> ### Author Rebuttal · Authors · 2025-07-31
>
> We would like to thank the reviewer for the thorough reading of our work and the positive feedback regarding RICOL. The reviewer has provided positive feedback on our overall framework, referring to it as a "novel and interesting" idea that merges Retrospective In-Context Learning (RICL) with Advantage-Weighted Regression to form a full online RL loop. The reviewer has raised several concerns. While we acknowledge some of these points, we respectfully disagree with others.
>
> > Q1: The empirical study demonstrates effectiveness in structured environments such as BabyAI but lacks evaluation in more complex, real-world scenarios.
>
> We added an additional experiment on two scenarios from the Baba Is AI benchmark [1], a grid-based puzzle environment inspired by a video game, Baba Is You. This benchmark is designed to test agents' reasoning and planning abilities in dynamic environments where rules can be rearranged during an episode. Each episode lasts 100 steps, and agents must adapt with dynamic environments by combining previously seen rules in novel ways—making it a challenging testbed for evaluating reasoning capabilities of LLM agents on long horizon tasks.
>
> Scenario: `goto_win`
> | Algo Name    | Max Succ. Rate (%) | First Step That Achieves Max Succ. Rate | Succ. Rate at Time Step 7680 (%) |
> | ------------ | -------------- | --------------------------------------- | ---------------------------- |
> | RICOL (ours) | 100.0 ± 0.0        | 7680                                    | 100.0 ± 0.0                      |
> | PPO (3B)     | 100.0 ± 0.0        | 87040                                   | 43.8 ± 4.4                   |
>
> Scenario: `goto_win-distr_obj`
> | Algo Name    | Max Succ. Rate (%) | First Step That Achieves Max Succ. Rate | Succ. Rate at Time Step 14848 (%) |
> | ------------ | -------------- | --------------------------------------- | ---------------------------- |
> | RICOL (ours) | 95.3 ± 1.9     | 14848                                   | 95.3 ± 1.9                   |
> | PPO (3B)     | 100.0 ± 0.0        | 107520                                  | 28.9 ± 2.0                   |
>
> Results show that RICOL achieves comparable performance to PPO (3B) on these scenarios while being ~10 times more sample efficient, highlighting its effectiveness on more complex, reasoning-intensive tasks.
>
> > Q2: The proposed methods need an additional reflector LLM and a second forward pass of the policy, which may introduce significant computational overhead. Can you elaborate more on the computational overhead of the RICL? Will it take much longer inference time than the other methods mentioned in the paper?
>
> To provide a fair comparison, we have reported the total wall-clock training time (in minutes) in Appendix D.
>
> | Method     | goto | pickup | pick_up_seq_go_to | open | GPU Type                   |
> |------------|------|--------|--------------------|------|----------------------------|
> | RWR        | 72   | 69     | 232                | 227  | NVIDIA A40 × 2            |
> | RICOL      | 96   | 95     | 594                | 585  | NVIDIA A40 × 2            |
> | PPO (3B)   | 1440 | 1440   | 1440         | 1440 | NVIDIA A40 × 4            |
> | PPO (10M)  | 258  | 257    | 234            | 198  | GeForce RTX 2080 Ti × 1   |
>
>
> Our results show that RICOL requires approximately 1.5× to 2× more training time than the RWR baseline. This overhead primarily stems from the step-wise reflection process. However, RICOL is significantly more efficient than PPO (3B), as it converges in far fewer environment steps, thereby saving substantial policy inference time, offsetting the reflection cost. We also want to emphasize that in many real-world scenarios, such as robotics or scientific discovery, simulation or data collection can be expensive. In such cases, sample efficiency (i.e., requiring fewer environment steps) is often a more critical metric than token efficiency. By enabling faster policy adaptation with fewer samples, our method offers practical benefits in these domains.
>
>
> > Q3: The paper compares the RIOCL with the PPO(3B) in Figure 4, which seems to be unfair. The base model should have the same size as the RIOCL. Can you provide the PPO(7B) model results to have a fairer comparison with the RICL in BabyAI?
>
> The comparison in Figure 4 is fair, as RICOL also uses a 3B model as the backbone. (Please check line 298.) To ensure consistency, we align key components between RICOL and PPO (3B), including the prompt template and the policy model size. This setup allows for a direct and meaningful comparison of the learning algorithms.
>
> > Q4: Can you elaborate more on the difference between the RICL and the previous methods mentioned in the paper, such as the Reflexion and RICO-GRPO?
>
> RICOL vs. RICO-GRPO: Both methods avoid learning an explicit value function. However, RICO-GRPO utilizes Monte Carlo return estimation, alongside group normalization for approximating the value function and computing advantages. This sample-based approach requires higher computation cost to collect multiple rollouts and can lead to high estimation variance, when applied in long-horizon, multi-turn agentic tasks. On the other hand, RICOL performs credit assignment by directly inferring the advantage function through in-context updated and current policies, as shown in Eq. 3. In section 6.2, we empirically show that RICL needs 100x less samples to obtain a decent advantage function estimation, compared with Monte-Carlo-based methods.
>
> RICL vs. Reflexion: RICL generates verbal feedback for each individual action, whereas Reflexion provides a single, trajectory-level feedback. As a result, Reflexion’s feedback is coarser. Moreover, RICL updates the policy in-context for the specific states it recently encountered, eliminating the need to generalize to unseen states. In contrast, Reflexion relies on broader generalization, which can be less reliable. In Section 6.3 of the main paper, we compare the in-context updated policies produced by RICL and Reflexion, and show that RICL yields a superior policy—closer to the optimal policy—on the BabyAI task.
>
> > Q5: Can you explain why the open task performs worse than the GPT-4o-mini while the other three tasks perform better than the GPT-4o-mini in Figure 4? Is there any specific reason for this observation?
>
> The relatively poor performance of RICOL on the open task is primarily due to the low zero-shot performance of the underlying policy, i.e., Llama-3.2-3B-Instruct. The agent frequently gets stuck, either by repeatedly bumping into walls or endlessly turning left and right, resulting in uninformative trajectories. These suboptimal behaviors provide little learning signal for the reflector, making it difficult to improve the policy through reflection. In contrast, larger models like GPT-4o-mini demonstrate stronger initial performance on this task and are less prone to such failure modes. GPT-4o-mini is not an open-source LLM, otherwise we can use it as an underlying policy and get a better performance on the open task.
>
> [1] Cloos, Nathan, et al. "Baba is AI: Break the rules to beat the benchmark." arXiv preprint arXiv:2407.13729 (2024).

---

> > ### Comment · Area_Chair_bX1p · 2025-08-04
> > **Please respond to the author's rebuttal post**
> >
> > Hi Reviewer b3Y6, I see no response letting me know whether or not the rebuttal has changed your opinion. Could you please let me and the authors know by engaging? This process is critical to enabling the (S)ACs to make a decision on this work.
> >
> > --Your AC

---

> > ### Comment · Reviewer_b3Y6 · 2025-08-07
> >
> > Thank you for the detailed response. It addresses my concerns. I will raise my score.

---

> > > ### Author Response · Authors · 2025-08-08
> > >
> > > Thank you very much!

---

### Official Review · Reviewer_uB1k · 2025-07-05

**Clarity:** 4
**Significance:** 4
**Originality:** 4
**Rating:** 5
**Confidence:** 2

**Summary:**

This paper presents a novel Retrospective In-Context Learning (RICL) method and an associated online reinforcement learning framework, Retrospective In-Context Online Learning (RICOL), aimed at improving temporal credit assignment for LLM-based agents in sparse reward environments. The key insight is to leverage a Reflector LLM to generate verbal feedback based on observed trajectories, which is retrospectively incorporated into the actor LLM's prompt to update its behavior in a trajectory-specific manner. By comparing the log-probabilities of actions before and after in-context updates, the method estimates advantage functions, converting sparse feedback into dense supervision signals.

**Questions:**

The effectiveness of RICL hinges on the quality of verbal feedback generated by the Reflector LLM, yet:
1. Does the Reflector require task-specific prompt engineering or fine-tuning?
2. Have the authors tested RICOL with intentionally flawed or biased Reflector outputs to assess worst-case robustness?

**Ethical Concerns:**

["NO or VERY MINOR ethics concerns only"]

**Final Justification:**

Dear AC,

I have discussed with the author and appreciate their detailed rebuttal.

This paper is solid and addresses an important issue of credit assignment in RL for LLM agents. The proposed Retrospective In-Context Learning method provides a promising solution.

**Limitations:**

The same as the weaknesses.

**Paper Formatting Concerns:**

No formatting issue exists.

**Quality:**

4

**Strengths And Weaknesses:**

Strengths:
1. The proposed use of advantage estimation is grounded and offers an elegant alternative to traditional Monte Carlo or value-based methods, particularly under sparse rewards.
2. The Reflector LLM provides fine-grained, state-specific verbal feedback, addressing limitations of prior trajectory-level reflection approaches like Reflexion.
3. Experimental results are comprehensive, spanning interpretable synthetic tasks and more challenging BabyAI environments, with consistent improvements in sample efficiency and task success rates.

Weaknesses:
1. The approach depends on the Reflector LLM generating high-quality, task-relevant feedback, and may degrade under systematic biases or poorly designed prompts.
2. Theoretical guarantees are contingent on idealized assumptions regarding environment stationarity and policy-reward consistency, which may not hold universally.

---

> ### Author Rebuttal · Authors · 2025-07-31
>
> We thank the reviewer for the thoughtful feedback. We appreciate the recognition that RICL offers an elegant alternative to value-based advantage estimation, and that our experiments are comprehensive across synthetic and BabyAI tasks. Below, we address the reviewer’s concerns.
>
> > Q1: The approach depends on the Reflector LLM generating high-quality, task-relevant feedback, and may degrade under systematic biases or poorly designed prompts.
>
> We acknowledge that the performance of RICOL heavily depends on the quality of the verbal feedback generated by the Reflector LLM. If the prompt is poorly designed or the trajectory lacks informative reward signals, the effectiveness of RICOL can degrade. However, we note that in our experiments, we used a single, fixed prompt across all BabyAI tasks, as shown in Fig. 11 and Fig. 12 in the appendix. This prompt was not tailored to any specific scenario, suggesting that RICOL is reasonably robust even without task-specific prompt tuning.
>
> > Q2: Theoretical guarantees are contingent on idealized assumptions regarding environment stationarity and policy-reward consistency, which may not hold universally.
>
> In Theorem 4.1, we assume finite state and action spaces, as well as environment stationarity, conditions typically satisfied in the token MDP [1] setting of LLMs.
>
> > Q3: Does the Reflector require task-specific prompt engineering or fine-tuning?
>
> We use the same prompt template across all BabyAI scenarios, as shown in Figure 12 of the appendix. To further test the generality of our approach, we conducted additional experiments during the rebuttal period on two BabaIsAI scenarios [2]. In these experiments, we reused the same prompt template, modifying only the system prompt to reflect the rules of the BabaIsAI environment. The results, shown below, indicate that minimal prompt engineering is needed for the Reflector to function effectively.
>
> Scenario: `goto_win`
> | Algo Name    | Max Succ. Rate (%) | First Step That Achieves Max Succ. Rate | Succ. Rate at Time Step 7680 (%) |
> | ------------ | -------------- | --------------------------------------- | ---------------------------- |
> | RICOL (ours) | 100.0 ± 0.0        | 7680                                    | 100.0 ± 0.0                      |
> | PPO (3B)     | 100.0 ± 0.0        | 87040                                   | 43.8 ± 4.4                   |
>
> Scenario: `goto_win-distr_obj`
>
> | Algo Name    | Max Succ. Rate (%) | First Step That Achieves Max Succ. Rate | Succ. Rate at Time Step 14848 (%) |
> | ------------ | -------------- | --------------------------------------- | ---------------------------- |
> | RICOL (ours) | 95.3 ± 1.9     | 14848                                   | 95.3 ± 1.9                   |
> | PPO (3B)     | 100.0 ± 0.0        | 107520                                  | 28.9 ± 2.0                   |
>
> Results show that RICOL achieves comparable performance to PPO (3B) on these scenarios while being ~10 times more sample efficient.
>
> > Q4: Have the authors tested RICOL with intentionally flawed or biased Reflector outputs to assess worst-case robustness?
>
> Yes. Section 6.6 evaluates RICOL’s robustness using intentionally flawed, hand-crafted verbal feedback in the BabyAI goto task. The feedback from the Reflector is randomly flipped to simulate biased Reflector outputs. RICOL remains effective even with 70% accurate feedback, demonstrating resilience to noisy supervision. We attribute this robustness to the trust region term in the Advantage-Weighted Regression (AWR) style policy update, as shown in Eq. 5, which stabilizes learning under imperfect feedback.
>
> [1] Rafailov, Rafael, et al. "Direct preference optimization: Your language model is secretly a reward model." Advances in neural information processing systems 36 (2023): 53728-53741.
>
> [2] Cloos, Nathan, et al. "Baba is AI: Break the rules to beat the benchmark." arXiv preprint arXiv:2407.13729 (2024).

---

> > ### Comment · Area_Chair_bX1p · 2025-08-04
> > **Please respond to the author's rebuttal post**
> >
> > Hi Reviewer uB1k, I see no response letting me know whether or not the rebuttal has changed your opinion. Could you please let me and the authors know by engaging? This process is critical to enabling the (S)ACs to make a decision on this work.
> >
> > --Your AC

---

### Decision · Program_Chairs · 2025-09-17

**Decision:**

Accept (poster)

**Comment:**

The paper presents a method to transform sparse rewards into relatively dense advantages by using the differences between the log probs of the original policy as well as an in-context learned version. The idea itself seems novel and interesting, with the majority of reviewers agreeing that it is a potentially useful avenue to build on. While there are concerns about the limitations of the eval setting (BabyAI being quite limited) and the robustness of the methods to various prompt templates for in context learning, overall the merits definitely outweigh the limitations and warrant publication.